# Data assimilation of SMOS observations into the Mercator Ocean operational system: focus on the Niño 2015 event

Benoît Tranchant[1], Elisabeth Remy[2], Eric Greiner[1] and Olivier Legalloudec[2]

[1]Collecte Localisation Satellites, Ramonville Saint-Agne, 31520, France

[2]Mercator-Océan, Ramonville Saint-Agne, 31520, France

*Correspondence to*: Benoît Tranchant (btranchant@groupcls.com)

**Abstract.** Monitoring **S**ea **S**urface **S**alinity (SSS) is important for understanding and forecasting the ocean circulation. It is even crucial in the context of the intensification of the water cycle. Until recently, SSS was one of the less observed essential ocean variables. Only sparse in situ observations, mostly closer to 5 meters depth than the surface, were available to estimate

the SSS. The recent satellite ESA Soil Moisture and Ocean Salinity (SMOS), NASA Aquarius SAC-D and Soil Moisture Active Passive (SMAP) missions have made possible for the first time to measure SSS from space and can bring a valuable additional constraint to control the model salinity. Nevertheless, satellite SSS still contain some residual biases that must be removed prior to bias correction and data assimilation. One of the major challenges of this study is to estimate the SSS bias and a suitable observation error for the data assimilation system. It was made possible by modifying a 3D-Var bias correction

scheme and by using the analysis of the residuals and errors with an adapted statistical technique.

This article presents the design and the analysis of an Observing System Experiment (OSE) conducted with the ¼° resolution Mercator Ocean global analysis and forecasting system during the El-Niño 2015/16 event. The SSS data assimilation constrains the model to be closer to the near-surface salinity observations in a coherent way with the other datasets already routinely assimilated in an operational context. This also shows that the overestimation of E-P is corrected by data assimilation through

salting in regions where precipitations are higher. Globally, the SMOS SSS assimilation has a positive impact in salinity over the top 30 meters. Comparisons to independent salinity data sets show a small but positive impact and corroborate the fact that the impact of SMOS SSS assimilation is larger in the ITCZ and SPCZ regions. There is little impact on the Sea Surface Temperature (SST) and Sea Surface Height (SSH) error statistics. Nevertheless, the SSH seems to be impacted by the Tropical Instability Waves (TIWs) propagation, itself linked to changes in Barrier Layer Thickness (BLT).

Finally, this study helped us to progress in the understanding of the biases and errors that can degrade the SMOS SSS data assimilation performance.

## 1 Introduction

Recent progress in data treatment of Sea Surface Salinity (SSS) from space makes possible their assimilation in ocean analysis systems (Boutin et al., 2017). Since the launch of the European Space Agency (ESA) Soil Moisture and Ocean Salinity (SMOS) mission in 2009, then the launchs of NASA'a Aquarius in 2011 and Soil Moisture Active Passive (SMAP) in 2015, SSS observations from space are available and have been used in many studies (e.g., Tang et al., 2017, Vinogradova et al., 2014; Toyoda et al., 2015, Reul et al., 2013).

Here we present the impact of assimilating SSS observations from space into the global ¼° Mercator Ocean operational system (see Lellouche et al., 2013) evaluated in the SMOS Niño 2015 project (https://www.godae-oceanview.org/projects/smos-Niño15). The changes induced by assimilating the satellite SSS in addition to the observation data operationally assimilated are analyzed. The focus has been primarily on the 2015-2016 El Niño event, in which strong SSS anomalies are seen in the Tropical Pacific in both model and observations (Hasson et al., 2018; Gasparin and Roemmich, 2016; Guimbard et al., 2017). The salinity plays an important role in the ocean-atmosphere coupling in this region by isolating the ocean interior due to the formation of a barrier layer. It is then not only the thermocline depth that is of importance but also the halocline when it becomes shallower than the thermocline.

The most striking event in the global ocean for the year 2015 was the strong El Niño event. It is as strong as in 1997(Von Schuckmann et al., 2018). Because the maximum of the SST anomalies stays off the eastern coast of South and Central America, it was more likely to be a El Niño Modoki (Ashok and Yamagata, 2009) or a central Pacific El Niño (Kao and Yu, 2009) than a classical eastern Pacific El Niño.

Warm anomalies began to build in the western pacific in 2014 triggered by Westerly Wind Bursts but did not lead to the development of an El Niño in the year. However, as suggested by McPhaden et al., (2015), the presence of El Niño precursors in early-2014 helped the development of a strong El Niño at the end of 2015. Anomalously eastward currents along the equator and in the NECC continued from 2014. This is associated with an increase in precipitation and an eastward shift in fresh surface salinities. A strong equatorial SSS anomaly in 2015 has been observed and described (Hasson et al., 2018; Gasparin and Roemmich, 2016). The Pacific freshening is due to an active ITCZ in 2015, but advection by anomalous eastward currents also plays a role in the SSS changes. The difference of the two annual SSS anomalies in 2014 and 2015 in our so-called Reference simulation (hereafter REF) (see section 3) is shown in Figure **1**. The 2015–2016 El Niño is also the first important climatic event fully captured by the SMOS satellite where negative SSS anomalies have been observed between 0 and 15°N around 170°W from mid-2014 to mid-2015 (Boutin et al., 2016).

Data assimilation experiments conducted within the SMOS Niño 2015 project (https://www.godae-oceanview.org/projects/smos-Niño15) are helping to prepare the assimilation of space SSS data and allow testing their impact on short term ocean forecast and analysis. To evaluate the impact of SSS observations from satellites on ocean monitoring and forecast systems in a realistic context, Observing System Experiments (OSEs) were conducted with the UK Met Office and Mercator Ocean global ocean forecast systems. Two simulations are compared, one with and the other without SSS data

assimilation. The differences between the two simulations highlight the "impact" of the withheld observations. Similar OSE approaches are generally used to evaluate observation networks in the ocean data assimilation community of GODAE OceanView (Oke et al., 2015, Lea et al., 2014).

Experiments conducted within the SMOS Niño15 project to test the impact of the satellite SSS data were carefully designed and analyzed to ensure robust conclusions on the impact of SSS measurements on ocean analysis. The system used for the OSE is based on the operational ocean monitoring and forecasting system operated at Mercator Ocean. The use of such system ensures that conclusions are relevant for operational applications.

To assess the benefit of assimilating SSS from satellite in a realistic context, all observations from the Global Ocean Observing System (GOOS) that are assimilated in real time ocean analysis or reanalysis are also assimilated. SST, in-situ temperature and salinity observations (from moorings, drifting platforms, ships) and along track Sea Level Anomalies are assimilated in the REF simulation and OSEs. OSEs conducted were designed to assess the impact of weekly SSS products as the system has a weekly assimilation cycle.

It is recommended to withhold part of the usually assimilated observations from the OSEs to have fully independent data to compare with, see (Fuji et al., 2015). The TAO mooring salinity data were not assimilated and kept for verification. Although restricted to the few mooring points, those data are the only ones to provide long term time series of daily temperature and salinity observations.

Several studies (Reul et al., 2013 or Lee et al., 2012) show that SSS measured from space can bring new information. Recently, (Toyoda et al., 2014; Hackert et al., 2014) show the impact of assimilating Aquarius data in the Pacific region both in uncoupled and coupled ocean-atmosphere systems. In a recent paper, Chakraborty et al., (2014) show that the migration of the thermohaline fronts at the eastern edge of the western Pacific warm pool can be more realistic with the assimilation of Aquarius SSS. Data assimilation of Aquarius SSS can also help to better understand the variability of salinity structure in the Bay of Bengal (Seelanki et al., 2018). Finally, satellite SSS data assimilation is promising in an operational context both for ocean and seasonal forecasting.

Nevertheless, technical challenges are still open to assimilate SSS data efficiently in the context of global ocean analysis and forecasting. The assimilation of satellite SSS observations is challenging because of the various complex biases, see Köhl et al, (2014). The difference between the forecast and the satellite SSS can be 5 times larger than the misfit between the forecast and near surface ARGO salinity. Since the signal to noise ratio is still not high today, retrieval algorithms must be improved. Careful analysis of the SSS data sets shows that a bias correction is needed before their assimilation as shown by Martin (2016). To have an optimal analysis, the hypothesis of un-biased errors has to be respected. This article details the bias correction scheme and the error estimation scheme used in the data assimilation system for those data. This is a necessary step to have a positive impact of SSS data assimilation.

The structure of this article is as follows: after a description of the OSE where the operational system, the bias correction, the SSS observation error and the presentation of the experimental design are described in section 2, the effect of the SMOS SSS data assimilation is presented in section 3, while discussions and conclusions are provided in section 4.

## 2. OSE approach

The OSE are conducted with the global ¼° ocean analysis and forecasting system running in real time at Mercator Ocean. Detailed descriptions of the system can be found in (Lellouche et al., 2013; Lellouche et al., 2018). After a brief description of the system configuration, we will describe the data assimilation components that were specifically developed or adapted for the SSS data assimilation in details.

### 2.1 Ocean model and configuration

The Mercator Ocean real time analysis and forecast is based on the version 3.1 of the NEMO ocean model (Madec, 2016), which uses a ¼° ORCA grid. The water column is discretized into 50 vertical levels, including 22 levels within the upper 100 m, with 1-m resolution at the surface to 450-m resolution at the bottom. The system has been initialized in autumn 2006, using temperature and salinity profiles from the EN4 climatology (Good et al., 2013).

The ocean model is forced by atmospheric fields from the European Centre for Medium-Range Weather Forecasts-Integrated Forecast System (ECMWF-IFS) at 3-hr resolution to reproduce the diurnal cycle. Momentum and heat turbulent surface fluxes are computed by using (Large and Yeager 2009) bulk formulae. Because there are large known biases in precipitation, a satellite-based large-scale correction of precipitation is applied to the precipitation fluxes. This correction has been inferred from the comparison between the Remote Sensing Systems (RSS) Passive Microwave Water Cycle (PMWC) product (Hilburn, 2009) and the IFS ECMWF precipitation (Lellouche et al., 2013).

A monthly river runoff climatology is built with data on coastal runoff from 100 major rivers from the Dai et al. (2009). This database uses new data, mostly from recent years, streamflow simulated by the Community Land Model version 3 (Verstentein et al., 2004) to fill the gaps, in all land areas except Antarctica and Greenland. At high latitudes the effect of iceberg melting is also parameterized. The lack of interannual variability of the largest rivers is known to lead to large errors in the surface ocean salinity in the analysis and forecast. There is no SSS relaxation term to any climatology as it is the case in operational conditions. More details concerning parameterization of the terms included in the momentum, heat and freshwater balances (i.e, advection, diffusion, mixing and surface fluxes) can be found in Lellouche et al., (2018).

### 2.2 Assimilated Observations

#### 2.2.1 Regular observation data

All ocean observations assimilated in the real time forecasting system are assimilated in the same way in the OSEs presented here. Along track SLA observations distributed by CMEMS (http://marine.copernicus.eu/) referenced to an unbiased Mean Dynamic Topography (MDT) based on the CNES/CLS 2013 MDT are used. Gridded satellite SST OSTIA Level 4 (L4: SST analysis using optimal interpolation (OI) on a global 0.054 degree grid) are assimilated each week in addition to SST measurements from the in-situ database delivered by the CORIOLIS centre (http://www.coriolis.eu.org/). Assimilation of in situ temperature and salinity profiles from this database are mostly from Argo floats, XBT, CTDs, moorings, gliders and sea

mammals. The assimilation of those routine observations in the OSEs provides a realistic context for the global ocean observing system so that the experiments address the complementarity of the different data sets with satellite SSS. The only exception is the TAO mooring observations of salinity that are withheld from the analysis and kept as independent observations to evaluate the performance of the assimilation experiment and the impact of the SSS assimilation. The model SSS in the real time system

is only constrained at large scale by in-situ observations, mostly Argo floats that usually start to measure at 5 meters depth.

### 2.2.2 SSS from space

In this study, we assimilate a SMOS Level 3 (L3: provided on a grid, but with no in-filling) SSS product at 0.25° resolution. L3 products are qualified (quality controlled) and processed at the Data Production Center (CPDC) of the Centre Aval de Traitement des Données SMOS (CATDS CEC-LOCEAN) (Boutin et al., 2017). Compared to Level 2 products (L2: SSS

values at the native swath resolution), they benefit from additional corrections. These are 18-day products sampled at 25km resolution provided every 4 days (the precise description of the time filtering is in the documentation at http://www.catds.fr/Products/Available-products-from-CEC-OS/L3-Debiased-Locean-v2). We checked that this temporal resolution fits well the model resolution and the weekly analysis window used in the assimilation scheme, see next section. In practice, the gridded SSS which is the closest to the analysis date (i.e the fourth day of the week) provides the SSS data for the

cycle. The model counterpart is the time average over the weekly cycle. Due to a low signal to noise ratio, the assimilation of the SSS data is limited in the latitudinal band between 40°S and 40°N.

### 2.3 Data Assimilation Scheme

The assimilation scheme implemented in the real time Mercator Ocean systems is based on a reduced order Kalman Filter called SAM2 (Système d'Assimilation Mercator V2) and is described in Lellouche et al., (2013) and Lellouche et al., (2018).

As in the operational ocean forecasting system, we use a weekly assimilation cycle with an analysis date on the fourth day of the week.

### 2.3.1 Background Error Covariances

The SAM2 system uses a background error covariance matrix based on a reduced basis of a fixed collection of multivariate model anomalies. The model anomalies are computed from a previous simulation over a 8 year period with an in-situ bias

correction, detailed in the section 2.4. The forecast error covariances rely on a fixed basis, seasonally variable ensemble of anomalies calculated from this long experiment. A significant number of anomalies are kept from one analysis to the other, thus ensuring error covariance continuity. The aim is to obtain an ensemble of anomalies representative of the error covariance (Oke et al., 2008), which provide an estimate of the error on the ocean state at a given period of the year. The localization of the error covariance is performed assuming a zero-covariance beyond a distance defined as twice the local spatial correlation

scale (Lellouche et al., 2013). These spatial correlation scales are also used to select the data around the analysis point. The

model correction (analysis increment) is a linear combination of these anomalies. This correction is applied incrementally over the assimilation cycle temporal window using an incremental analysis update (Bloom et al., 1996; Benkiran and Greiner 2008).

### 2.3.2 Observation Error Covariances

The observation errors specified in the assimilation scheme are assumed to be uncorrelated with each other. Observation errors
include representativity errors specified as a fixed error map and an instrumental error. Representativity errors for in situ observations were calculated a-posteriori from a reanalysis over the period 2008-2012. The applied statistic method (Desroziers et al., 2005) consists of the computation of a ratio, which is a function of observation errors, innovations and residuals. These estimated errors are constant throughout the year.

The instrumental errors of SLA, SST and in situ measurements are summarized in Table 1. Figure 2a shows the representativity
error used for the in-situ SSS and an example of the resulting salinity error (Figure 2b) for in-situ data for the week 20-27 January 2016. The SSS error from space is estimated during the bias correction scheme procedure (see section 2.5) and then used in SAM2.

### 2.4 Bias correction scheme

### 2.4.1 Bias correction scheme for large scale 3D temperature and salinity: in-situ T/S

Biases between model and data exist for subsurface quantities such as temperature and salinity. As with the time-varying error components, such biases can often be related to systematic errors in the forcing (Leeuwenburgh, 2007).

As written in Lellouche et al., (2013), a 3D-Var bias correction is applied for large scale 3D temperature and salinity fields. The aim of this bias correction is to correct the large-scale, slowly evolving errors of the model, whereas the SAM assimilation scheme is used to correct the smaller scales of the model forecast error.

This is applied separately to the model's prognostic T/S equations from in-situ profile innovations calculated over the preceding month on a coarse grid (1°x1°). This bias is the minimizer of the cost function given by the Eq. 1.

$$J(x) = \frac{1}{2} x^T B^{-1} x + \frac{1}{2} (d-H x)^T R^{-1} (d-H x) \tag{1}$$

where $d = $ <Salinity$_{in-situ}$> - <Salinity$_{model}$> for salinity field

Here, $d$ is the innovation vector of T/S, i.e the mean innovation of in-situ T/S over 1 month in a 1°x1° grid boxes. Salinity$_{in-situ}$ and Salinity$_{model}$ denote salinity values of in-situ data and model, and <·> indicates the mean.

$x$ is the temperature or salinity in-situ bias to estimate, $B$ denotes the background error covariance of the 3D bias, $d$ is the innovation vector, $H$ is the observation operator, $R$ is the observation covariance error. The vertical grid is a coarse grid (only 23 levels) which is different of the model vertical grid (50 levels). For example, the in-situ innovation at sea surface for T/S is calculated from the average of model and observations between 0 and 11meters depth.

Because temperature and salinity biases are not necessarily correlated at large scales, these two variables are processed separately. Spatial correlations in **B** are modeled by means of an anisotropic Gaussian recursive filter (Wu et al., 1992; Riishøjgaard, 1998; Purser et al., 2003). Finally, bias correction of T, S and dynamic height are computed and interpolated on the model grid and applied as tendencies in the model prognostic equations with a 1-month time scale.

### 2.4.2 Bias correction scheme for large scale SSS: SSS from space

Earlier attempts to assimilate SSS data have shown the importance of using unbiased satellite SSS data while implementing rigorous quality control in an upstream process (Tranchant et al., 2015). In this study, the bias control of satellite SSS has been modelled by modifying the current T/S bias (in-situ) correction 3D-Var cost function (Eq. 1). Two extra terms to take into account biases in the satellite SSS data has been added in the following 3D-Var cost function (Eq.2). The new SSS bias $\xi$ is the minimizer of the cost function given by the Eq. 2.

$$J(x,\xi)= \tfrac{1}{2} x^T \mathbf{B}^{-1} x + \tfrac{1}{2} (\mathrm{d}-\mathbf{H}\,x)^T \mathbf{R}^{-1} (\mathrm{d}-\mathbf{H}\,x) \tag{2}$$
$$+ \tfrac{1}{2} \xi^T B_\xi^{-1} \xi + \tfrac{1}{2} \left( d_\xi - \mathbf{H}_\xi\, x \right)^T R_\xi^{-1} \left( d_\xi - \mathbf{H}_\xi\, x \right)$$

where $d_\xi = (<\mathrm{SSS}_{SMOS}> - \xi) - <\mathrm{SSS}_{model(0.5m)}>$

Here, $d_\xi$ is the innovation of SSS bias at surface, i.e the mean innovation of satellite SSS over 1 month on a 1°x1° grid. $\mathrm{SSS}_{SMOS}$ denotes the original (non-debiased) SMOS SSS, $\mathrm{SSS}_{model(0.5m)}$ denotes the model SSS at 5 meters depth and the first term $(<\mathrm{SSS}_{SMOS}> - \xi)$ corresponds to the unbiased SMOS SSS. $\mathbf{H}_\xi$ is the linear operator which interpolates **x** to the positions of SMOS observations. $B_\xi$ denotes the background error covariance of the 2D satellite SSS bias and $R_\xi$ is the estimated SMOS SSS observation covariance error.

To get an optimal set of parameters (weights, spatial scales and errors), several estimations were performed with data withdrawing. Figure 3a and c show examples of the model salinity bias $x$, near the surface without (Eq.1) and with (Eq. 2) the estimation of the bias of SMOS data, $\xi$. The patterns are similar except at the equator where the estimated bias of SMOS data (Figure 3b) influences the estimated model salinity bias (Figure 3c) with smaller scales. In this example, a persistent large innovation at several depths (11m, 41 m and 79 m) (not shown here) induces a larger bias of salinity (negative anomaly) at sea surface near 120°W/20°S.

### 2.5 SSS observation error

The Desroziers diagnostic (Desroziers et al., 2005) is commonly used for estimating observation error statistics and is used here to adapt the observation error from the background and analysis residuals calculated in the bias correction (see also Lellouche et al., 2018). Following Desroziers et al., (2005), the observation error of the bias $R_\xi$ is optimal when is equal to the statistical expectation of the cross-product between the residual $d_\xi^a$ and the innovation $d_\xi$ of the SSS bias.

$$R_\xi = E[d_\xi . d_\xi^a] \tag{3}$$

Actually, $R_\xi$ is estimated iteratively (n=5) by an iterative boot-strap method computed on a 3°x3° grid. Five successive analyses are made followed by five estimates of the Desroziers ratio $r_\xi^i$ expressed as Eq. 4 for an analysis i.

$$r_\xi^i = \frac{E[d_\xi . d_\xi^{a_i}]}{R_\xi^{i-1}} \tag{4}$$

From an observation error a priori $R_\xi^0$ and by the successive ratio $r_\xi^{i=1,n}$, we obtain Eq.5:

$$R_\xi = r_\xi^n \ ... r_\xi^1 \ R_\xi^0 \tag{5}$$

The a priori error $R_\xi^0$ is a combination of a zonally varying error, together with an increase over regions with sparse in-situ data and near the coast. This increase varies with the cycle. It means that the SSS bias could not be estimated accurately in the absence of in situ data, and hence will have no impact in the assimilation in those regions void of in situ data. Figure 4 shows an example of the final Desroziers ratii product $r_\xi^5 r_\xi^4 r_\xi^3 r_\xi^2 r_\xi^1$. It illustrates how the fixed zonal error is increased near the equator and reinforced near central America where in situ data are sparse. There is also a local increase near Samoa (170°W-13°S), probably due to RFI pollution. Several simulations have been done with and without bias correction in order to check the validity of the estimated SSS errors in the data assimilation scheme SAM2.

Finally, for each weekly analysis, the total observation error of satellite SSS (SMOS) (Figure 5) prescribed in the data assimilation scheme is the maximum of the above observation error estimated during the bias correction process and the measurements error ($R_{instr.}$) supplied by the data producers (used as a threshold).

$$R_{Tot} = max(R_\xi , R_{instr.}) \tag{6}$$

These measurement error estimates bring smaller scales than can be estimated by the Desroziers diagnostic.

## 2.4 OSE design

Two parallel simulations were produced, the REF experiment and the SMOS experiment (hereafter SMOSexp) see Table 2. The only difference is the assimilation of the SSS SMOS observations. Both experiments begin in January 2014 from the same initial conditions coming from a previous reanalysis using only the bias correction of T/S without any data assimilation. The period covers the onset and development of the El-Niño 2015 event. The length of the OSE should at least cover one year, more if possible, as it takes 3 months for the system to be in equilibrium with the new data assimilated. This "adjustment" period is longer for observations deeper in the ocean (below the thermocline). Here, up to 2-year simulations are analyzed [January 2014 - March 2016].

The comparison between the two simulations highlights the impact of the SSS data assimilation on the ocean circulation and the comparison to the other observations (independent or not) will allow us to verify the coherency between the different observation networks and the way they are assimilated.

### 3. OSE analysis

Different diagnostics are now used to assess the impact of SSS data assimilation on the analysed model fields. First the analysis from the REF and SMOSexp simulations are evaluated against the assimilated observations. Then, the 3D fields of the simulations with and without SSS data assimilated are compared and the changes in the surface and subsurface fields are analysed. Finally, TAO/TRITON array salinity observations which are deliberately with-held and delayed time ThermoSalinoGraph (TSG) which are not assimilated in the analysis of all experiments are used to conduct an independent analysis-observation comparison. Our analysis focuses on the tropical Pacific region during the Niño 2015 event.

### 3.1 Assessment of the misfit reduction based on the data assimilated in the analysis

### 3.1.1 Assimilation diagnostics

The REF and SMOSexp simulations differ only by assimilating satellite SSS data (Table 2). We first check the success of the assimilation procedure in reducing the misfit from the assimilated SSS observations within the prescribed error bar. We then look at the Root-Mean-Square (RMS) of in-situ salinity observation innovations near 6 meters depth  in both simulations. The forecasted field is mostly independent of the reference data because those data have not been assimilated yet and the model forecast ranges  from1 to 7 days.

Figure 6 shows the time-series of Root-Mean-Square Errors (RMSEs) of the model near-surface salinity at 6 m depth with respect to in situ observations (dotted lines) and of the model SSS (0.5 m depth) with respect to the bias-corrected SMOS SSS (solid lines) for both simulations (REF in black, SMOSexp in red). As expected, the SMOS SSS data assimilation clearly leads to a significant reduction in the innovations of the SMOS data (solid lines). When the SSS SMOS is assimilated, the time series of RMSE for the global, the Tropical Pacific and the central Pacific (Niño3.4) domains present the same reduction with an higher variability for the smallest domain (Niño3.4). The global RMSE to SMOS data is around 0.28 pss (practical salinity scale) in the reference simulation and reduced to 0.21 pss when debiased SMOS data are assimilated, corresponding to an error reduction of 24%. This shows that the combination of bias correction and data assimilation perform well.

Nevertheless, the essential issue is the salinity RMSE compared to in-situ salinity observations (dotted lines). This error is slightly reduced from 0.20 pss to 0.19 pss in the global domain (5%), but this reduction can reach 10% in the Northern Tropical Pacific where the salinity anomaly is the strongest, see Table 3. This larger decrease in the near-surface salinity RMSE is consistent with that observed for the SSS SMOS RMSE (30%). In addition, the reduction of the near-surface salinity RMSE is more important in the western part of the Equatorial Pacific (Niño4). This shows that the assimilation of SMOS SSS observations does not introduce overall incoherent information and can even reduce the misfit to the in-situ salinity observations. It also confirms that SSS errors estimated in the bias correction procedure and used in the assimilation scheme are well tuned and the data bring coherent information. Consequently, salinity large scales biases are removed well. From Table 3, it should be mentioned that the number of in situ salinity observation per week is very small compared to the SMOS observations and maybe not always sufficient to ensure robust statistics in small regions.

Time series and maps of the misfits between observation and model forecasts are complementary to analyse the temporal and spatial variability of the model observation differences. Figure 7 shows the mean and root-mean-square differences of monthly mean SSS in the analysis fields in REF and SMOSexp compared to the original (non-debiased) SMOS data over the year 2015 for the Tropical Pacific Ocean.

5   The mean SSS bias in REF exhibits large scale patterns, coinciding with the 2015 SSS anomaly for the open ocean (Figure **1**). A large bias is also found in the Indonesian Archipelago. In contrast, the bias is effectively reduced in SMOSexp as well as the root-mean-square differences that is reduced to less than 0.2 pss (black isohaline) in most of the Tropical Pacific Ocean. The mean RMSE and the percentage of RMSE difference of the salinity profiles (mainly from Argo floats) are computed over the entire period and the global domain (Figure 8). There is a slight decrease in the first 30 meters below the surface when SSS 10   data are assimilated additionally to in-situ salinity data. It shows that the additional information brought by the SSS is in agreement with the salinity in-situ observations close to the surface. It can even help improving the global salinity representation in the first 30 meters by better constraining the model forecast with the satellite SSS.

In-situ temperature innovations in the global domain as well as in the Tropical Pacific region do not show significant changes. The same is found for SLA (SALTO/DUACS along track) and SST innovations (OSTIA L4). SSS data assimilation has a 15   quite-neutral impact on the innovations associated with those observations.

### 3.1.2 Impact of assimilating SMOS data during El-Niño 2015/16

We now look at the changes in the analysed surface and subsurface fields due to the SSS data assimilation by comparing the 3D analysis of the REF and SMOSexp experiments.  At basin scale, the REF simulation already agrees well with the 2015 mean deduced from the "unbiased" CATDS SMOS observations (Figure 9)**.** SMOS data assimilation induced changes in the 20   order of 0.2 pss. It tends to weaken the salinity negative anomaly represented in the REF simulation within the ITCZ and SPCZ regions. This is in agreement with Kidd et al., (2013) that show an overestimation of the ECMWF precipitation in the tropics compared to satellite observations. Elsewhere, the SMOS data assimilation increases the salinity. Large changes also occurred in the coastal zones (Indonesian archipelago and Central America coast), even if the specified error on SSS data was larger in those regions than in the open ocean.

25   The associated vertical salinity changes brought by SMOS SSS data assimilation at the equator are represented on Figure 10. The largest high-salinity anomaly are found in the first 50 m depth and along the coastal bathymetry, elsewhere changes are very small, less than 0.05 pss. Overall, at the equator (excepted in coastal areas), the data assimilation of SMOS SSS leads to fresher waters in the East and saltier waters in the West for the year 2015.

The highest variability of the surface salinity at monthly scale during the year 2015 is found within the ITCZ, SPCZ and in the 30   Eastern Pacific fresh pool, in both simulations and SMOS observations (not shown). SMOS assimilation decreases the intensity of the variability of the SSS, in agreement with the observed variability. In summary, the SSS assimilation acts to counteract the precipitation excess, with a visible result on the salinity both in terms of time mean but also in term of variability.

During the Niño2015 event, a strong salinity anomaly pattern developed in the Tropical Pacific (Gasparin et Roemmich 2016), see also Figure **1**. This anomaly corresponds to the ITCZ and SPCZ areas. Figure 11 shows the time-longitude evolution of the SSS at 5°N, the latitude where the salinity anomaly is the largest (Hackert et al., 2014). Both the REF and SMOSexp simulations represent the decrease of the salinity in fall 2015 between 160°E and 120°W. Note that this salinity anomaly is smaller in the SMOS data (SMOS SSS is saltier) with a smaller extent. The Eastern freshwater pool extended further west during 2015, but it was fresher in the REF simulation compared to the SMOSexp experiment.

While the impact of SSS assimilation is neutral on the other variables (temperature and SSH) in terms of data assimilation statistics (RMSE averaged in different areas), it is not the case when we look at the time evolution of model fields.

SST differences at 5°N and zonal velocity differences at the equator are represented on Figure 12. The differences are mainly associated with the wave propagation seen in all the surface fields. In the eastern freshwater pool, the SMOS data assimilation weakens the freshening and induces a slight warming of about 0.05°C (Figure 12b). At the equator, the zonal eastward advection is enhanced (positive pattern at the east of the date line) from January to October 2015 (Figure 12c) which could help the warm water pool migration to the East but this effect is very weak here. Note that the eastward warm water pool migration is known to promote the ocean-atmosphere coupling and thus the triggering of El Niño. In the Eastern basin, there is also an increase of the westward propagation during Autumn 2015 that are possibly linked to the increase of Tropical Instability Waves (TIWs) which will be shown later.

Another effect of SSS changes can be viewed on barrier layers which are quasi-permanent in the Tropical Pacific. Barrier Layer Thickness (BLT) can influence the air-sea interaction, ocean heat budget, climate change and onset of ENSO events, (Maes et al., 2002; Maes et al., 2004). The barrier layer acts as a barrier to turbulent mixing of cooler thermocline waters into mixed layer and thereby plays an important role in the ocean surface layer heat budget (Lukas and Lindstrom, 1991). The Hovmöller diagram of BLT at 5°N is shown on Figure 13 for both experiments. It shows the occurrence of thick BLT in the eastern Pacific ([120°W – 140°W]) from September to November which corresponds to measurements taken during strong El Niño events (Mignot et al., 2007). Note also that the eastward displacement of the thick barrier layer has already been observed during previous El Nino events (see.g., Qu et al., 2014).

From Figure 12a and Figure 13, we show that the Eastern and Central Pacific are saltier in the SMOSexp experiment which induce a decrease of the stratification and then a decreased BLT. A decrease of the stratification by SSS data assimilation can increase the convective mixing and the TIWs can be modified by this change of stratification. could be also enhanced by TIWs activity. From a long-term TAO mooring record at 0°N 140° W, Moum et al., (2009) suggest that mixing may always be enhanced during the passage of TIWs both in and below the surface mixed layer. Lien et al., (2008) show that turbulence mixing was modulated strongly by the TIW. Consequently, even if TIWs are less active during a El-Nino phase than in a La Nina phase, it was interesting to investigate the TIW propagation signature in SSH. Moreover, Yin et al., (2014) and Lee et al., (2012) show also the capability of monitoring TIWs by Aquarius and SMOS data. Lyman et al., (2007) show that TIWs, which have a 33-day period, are associated with the first meridional mode Rossby wave. Hovmöller of daily anomalies of SSH at 4°N filtered at 33 days are shown in Figure 14**.** For both experiments, the westward propagation of TIW is shown in the

Eastern part of the basin. A reinforcement of the TIWs at the eastern edge of the western Pacific warm pool near 140°W(the slope is steeper) appears during the end of the second half of 2015 in the SMOSexp experiment (0.35 m/s) compared to the REF experiment (0.20 m/s). As mentioned above, this could be correlated to the decrease of BLT(see Figure 13) which is associated to a mixing enhancement. On the contrary, a weakening of TIWs appears during the August-September period in the eastern part of the basin for the SMOSexp experiment. The same kind of impact have been shown recently in Hackert et al., (2014) for the initialization of the coupled forecast, where a positive impact of SSS assimilation is provided on surface layer density changes via Rossby waves. They also show that these density perturbations provide the background state to amplify equatorial Kelvin waves and ENSO signal.

## 3.2 Evaluation of the analysis toward independent observations

We now compare the analysed fields to independent observations, i.e. withheld from all assimilation experiments. This will allow verifying that the changes in the physical fields induced by the SMOS data assimilation are in agreement with external sources of information. For this purpose, the TAO mooring (salinity) observations and the reprocessed TSG data from the French SS Observation Service were withheld from all experiments. This is therefore a fully independent validation.

### 3.2.1 Comparisons to TAO mooring

TAO moorings deliver high frequency measurements at fixed locations. Such platforms allow us to look at high frequency variability that is not captured by drifting platforms. The hourly analysed salinity is collocated at the TAO mooring positions for the REF and SMOSexp simulations. Figure 15 shows the time evolution of TAO salinity observations (valid at 1 m depth) at three mooring locations in the equatorial Pacific (warm pool, cold tongue and salt front) compared to the model (analysis) for the REF and SMOSexp OSE experiments at the first level (~0.5 m depth). Assimilated SMOS data have also been added. In this example, the salinity evolution of the REF experiment (in green) appears less correlated with the TAO salinity mooring observations (black dots). The SMOSexp simulation shows a better agreement, except for some strongly variable events. The differences between the SMOSexp simulation and TAO non-assimilated observations are most of the time less than 0.1 pss. The high frequency variability seen in the observations is also reproduced in the assimilative simulations, with a better agreement when SMOS data are assimilated, except during some specific periods. Tang et al., (2017) also found some disagreement among the TAO, SMAP/SMOS, and Argo analysis during short periods. There is an improvement in the cold tongue during the end of summer, in fall 2015 and during the last 2 months of the SMOS simulation (Figure 15a). The data assimilation of SMOS reduces the freshening in this region. Globally, an improvement occurs also in the warm pool (15b) over the entire period. One interesting feature is that when TAO mooring data are missing during a long period near the salt front, the SSS from the SMOSexp experiment is different but closer to TAO mooring when measurements come back (Figure 15c). Obviously, the time-series of the assimilated SMOS data is smoother but are able to capture the large scales variability. This also shows the level of accuracy we need to capture higher variability. The precipitation rate superimposed on the SSS proves that it is not the only process that plays a role in the salinity variability. Indeed, a high precipitation rate does not induce

necessarily a strong freshening at the sea surface where advection, vertical mixing and SSS SMOS data assimilation can counteract its effect. This also shows that the observation error is not necessarily increased locally depending on the precipitation.

These three examples show a positive impact but it is also interesting to have a global view of all TAO moorings over the
2015/2016 El-Niño event. As in (Martin et al., 2018), Figure 16 shows the differences in RMSD from hourly TAO mooring salinity values at 1 m depth calculated over the period 1$^{st}$ Jan 2014 to 16$^{th}$ March 2016. The impact of the SMOS assimilation is contrasted by showing negative (positive) values which indicates that it reduces (increases) the RMSD. The impact is positive and more significant in the western Tropical Pacific near the dateline and in the western Pacific up to 5°N. The impact is quite neutral and even negative in the eastern tropical Pacific (140°W-110°W) between 2°S and 2°N where generally (i) the SMOS
bias is larger (Figure 3b), (ii) there are few in-situ SSS data (Figure 2) and (iii) where the observation error is larger (Figure 5). Actually, the impact of SMOS SSS assimilation is larger in the ITCZ and SPCZ regions as shown also in the Figure 9. This reflects the tendency that the SMOS data assimilation reduces the low salinity biases by mitigating the overestimation of E-P in the regions of large precipitation. Finally, during the El-Niño 2015/2016 event, there is a small positive impact overall from the SMOS assimilation with a reduction in RMSD from 0.326 to 0.316 pss (about 3%).

**3.2.2 Comparisons to ship SSS**

Post processed TSG observations from the French SSS Observation Service (SSS-OS; (http://www.legos.obs-mip.fr/observations/sss) were collected along the routes of voluntary merchant ships, see Alory et al., 2015. The SSS estimates have a ~2.5 km resolution along the ship track with an estimated error close to 0.08 pss. Salinity analysed fields from REF and SMOSexp simulations are collocated to the TSG observations. Salinity observations from vessel mounted thermosalinographs
allow validation of the short time and space scales of near surface salinity. Two ship routes (Figure 17a) that cross the Tropical Pacific Ocean in June 2015 are chosen to verify that salinity changes when SSS SMOS data are assimilated are in agreement with such observations.

Figure 17b and Figure 17c (zoom) show the comparison between the TSG salinity observations (in red) along the Matisse ship route collocated with the REF (black dashed line) and SMOSexp (black line) salinity analyzed fields. The variability of the
SSS measurements, lower than the daily frequency, is well represented in both simulations with only small differences of less than 0.2 pss except in the freshwater on the eastern part of the basin. In this region, the salinity dropped down to less than 34.0 pss. The REF simulation differs from the TSG data by more than 0.5 pss within the eastern freshwater pool, marked by a very sharp salinity front. The SMOSexp simulation shows a much better agreement with the SSS from the TSG observations: even if the differences remain large, the misfit is reduced. This confirms once again that the weakening of the freshening in the
freshwater pool in the eastern Pacific induced by the SMOS data assimilation is realistic, as it is seen by different in situ observation platforms.

## 4. Discussion and conclusions

The L3 SMOS CATDS data used in this study is considered as an "unbiased" product. Yet, they still contain some residual biases that must be removed prior to bias correction and data assimilation. One of the major challenge of this study was to estimate the residual SSS bias and a suitable observation error for the data assimilation system. It was made possible by using a 3D-Var bias correction scheme and an analysis of the residuals and errors with a statistical technique (Desroziers et al., 2005). The "debiased" data could then be assimilated by the SAM2 assimilation scheme which relies on the unbiased hypothesis. The bias estimated by the ocean forecasting system can also be used to correct the L3 SMOS CATDS data for other purposes.

The system was carefully tuned and tested to efficiently assimilate the new SSS observations before running the longer simulations that are analyzed here. The proper specification of the observation operator and error covariance matrix were also based on discussions with the data provider. This study helped us to progress in the understanding of the biases and errors that can degrade the SMOS SSS performance.

Nevertheless, there is still room for improvement. For instance, we used a zonal error as input to the error estimation with the Dezroziers technique. It could be beneficial to take into account the smaller scales linked to a shallow stratification that arises with strong precipitations and/or river runoff.

The SMOS data need accurate in-situ data (not only at the surface) to correct their own biases and estimate a suitable error (including data/system representativity). When enough accurate SMOS data are available, they really act as a gap filler. There is a clear impact on the scales about 1°-2°. This can be seen on the Figure 12 (Hovmöller), and additional spectral analyses (not shown) confirm this finding. So, it is important for future satellite SSS to provide a good accuracy at those scales. It also shows that background error correlation length-scales used in the bias correction scheme could be optimized with an improvement of the in-situ network and the SSS SMOS accuracy.

Globally, the SSS data assimilation slightly improves the simulation compared to a simulation assimilating only observations of in situ, SST and SLA data. It highlights that no incoherent information was brought by the SSS data compared to the other assimilated observations. When looking at the impact of the SMOS SSS assimilation, we found a positive impact in salinity with respect to in-situ data over the top 30 meters. The root mean square error (RMSE) of in-situ surface salinity is reduced in all regions of the Tropical Pacific and is very often close to 0.15 pss. The improvement varies depending on the region and can reach 10% in the North Tropical Pacific where the SSS anomaly is the strongest. Comparisons to independent TAO/Triton data corroborate the fact that the impact of SMOS SSS assimilation is larger in the ITCZ and SPCZ regions. This also reflects that the overestimation of E-P is mitigated by the data assimilation through salting in regions of large precipitations.

There is little impact on the SST. For instance, the area of the SST warmer than 28.5°C (warm pool region) was little affected. It means that the local impact on the air-sea coupling is negligible. But, an impact on TIW have been seen through SSH fields. Amplitude and propagation speed of TIWs are reduced while their activity is enhanced in the eastern part of the basin during the last half of 2015. This wave activity enhancement may induce a stronger mixing which decreases the BLT. Nevertheless,

the decreased BLT caused by an increase of sea surface salinity due to SMOS SSS assimilation may also enhance a stronger mixing. Another result can be seen on the strengthened Eastward advection of the warm pool in 2015 (Figure 12, Hovmöller of zonal velocity difference). These findings are close to those of Hackert et al., (2014) with a global ocean-atmosphere coupled model but benefits in term of seasonal forecasting have still to be quantified.

The next step will be to assimilate SSS from space at higher latitudes where low sea surface temperature (SST) degrades the brightness temperature sensitivity to SSS (Sabia et al., 2014). A longer ocean reanalysis with continuously improved SSS SMOS (available for over 9 years) and SMAP (available since 2015) data could bring new information on the water cycle. The focus of this study was on the tropical Pacific. But the system is global, and, in spite of RFI pollution near some coasts, we found clear improvements near the Amazon and the Rio Del Plata river plumes. So, the benefit from assimilating SMOS

SSS is not restricted to the equatorial band. Its positive impact near the mid-latitudes major rivers is a chance to better monitor the strengthening of the water cycle (Durack, 2015).

## Acknowledgements

We gratefully acknowledge funding from ESA as part of the SMOS-Niño15 project, coordinated by C. Donlon. We also thank the providers of the datasets used here. J. Boutin (LOCEAN/CATDS) provided the SMOS data and provided useful inputs to understand the nature of the SMOS bias estimates. Sea surface salinity data derived from voluntary observing ships were collected, validated, archived, and made freely available by the French Sea Surface Salinity Observation Service (http://www.legos.obs-mip.fr/observations/sss/). Thanks to the GTMBA Project Office of NOAA/PMEL to provide

TAO/TRITON mooring data. We would also like to acknowledge Matthew Martin (MetOffice) for his careful reading of the manuscript and his comments which were very helpful. We would also like to acknowledge the contribution of one anonymous reviewer whose suggestions improved this paper significantly.

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

**Figure 2: Representativity error of in-situ SSS ($R_{repr.}$) (a) and salinity error of in-situ data at sea surface (b) over the Tropical Pacific used in the data assimilation system for the week 20-27 January 2016.**

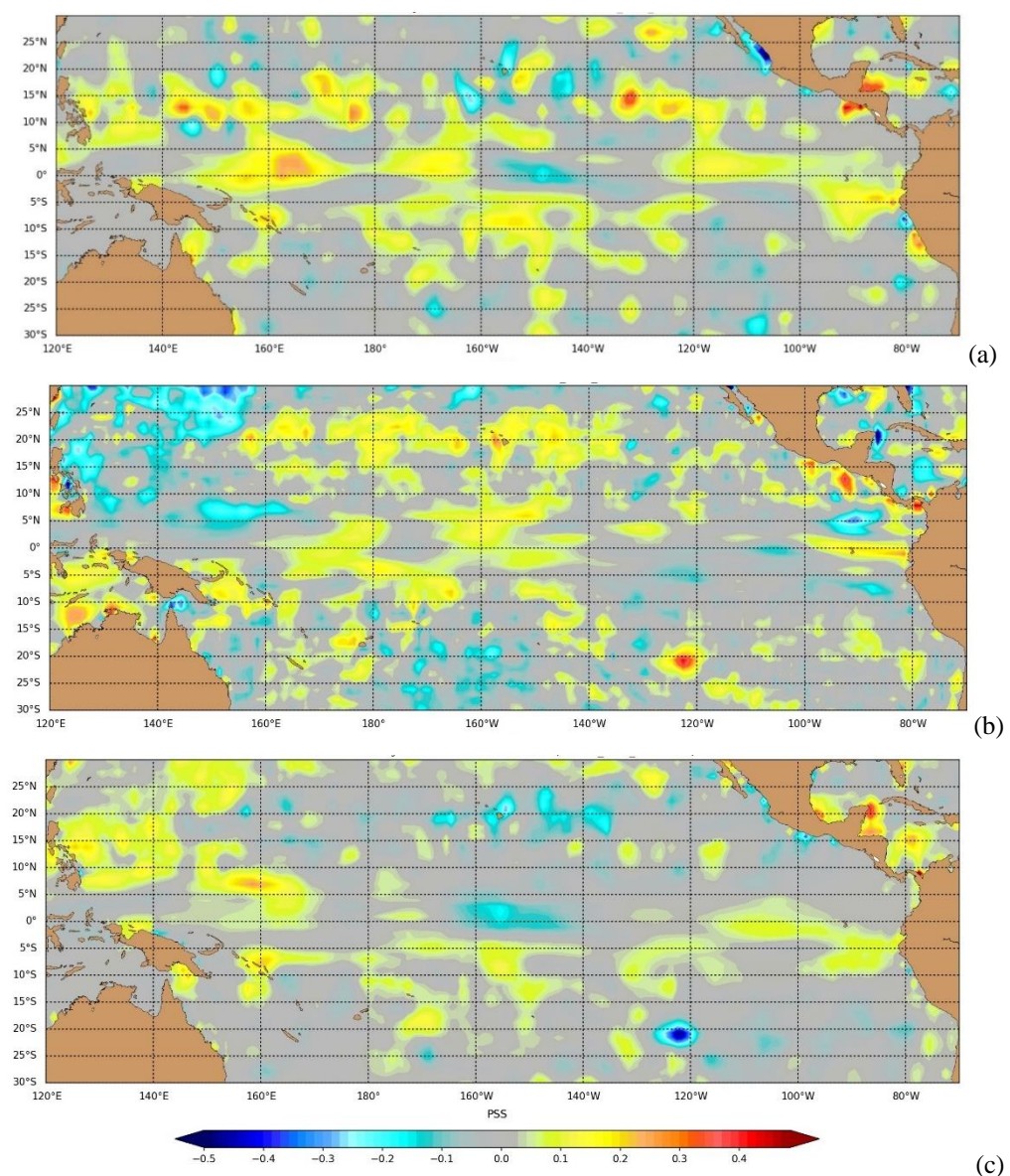

(a)

(b)

(c)

**Figure 3: Example of model salinity bias (x) near the surface (Eq. 1) calculated from in-situ data between 0 and 10 m depth only (a),**
5     **of SSS bias (ξ) (Eq. 2) calculated from SMOS SSS (b) and salinity bias (x) (Eq. 2) from in-situ data between 0 and 10 m and SMOS**
**SSS (c) in the Tropical Pacific (week 20-27 January 2016).**

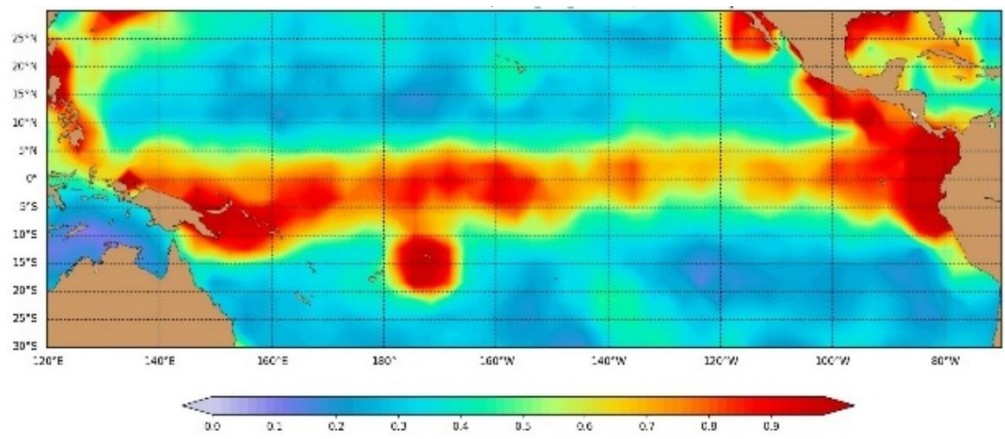

**Figure 4: Example of the final product of Desroziers ratii $r_\xi^5 r_\xi^4 r_\xi^3 r_\xi^2 r_\xi^1$ on a 3°X3° grid (see Eq. 4) estimated and applied to the a-priori error. (week 20-27 January 2016)**

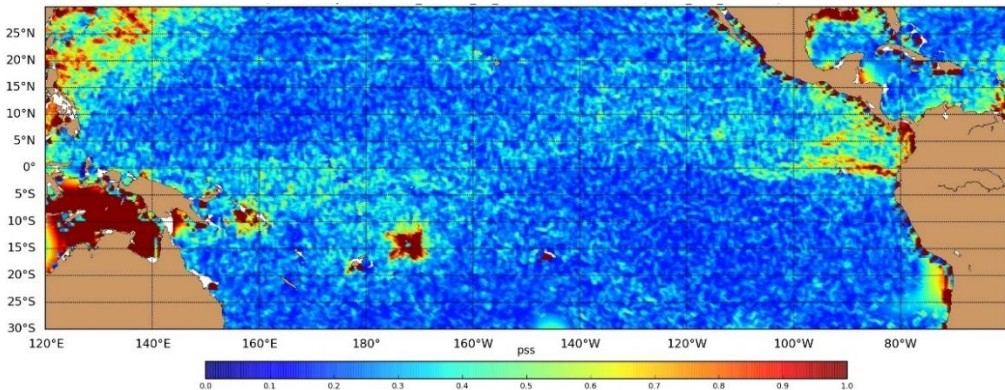

5   **Figure 5: Example of SSS error (Eq. 5) of SMOS over the Tropical Pacific and used in the data assimilation system for the week 20-27 January 2016.**

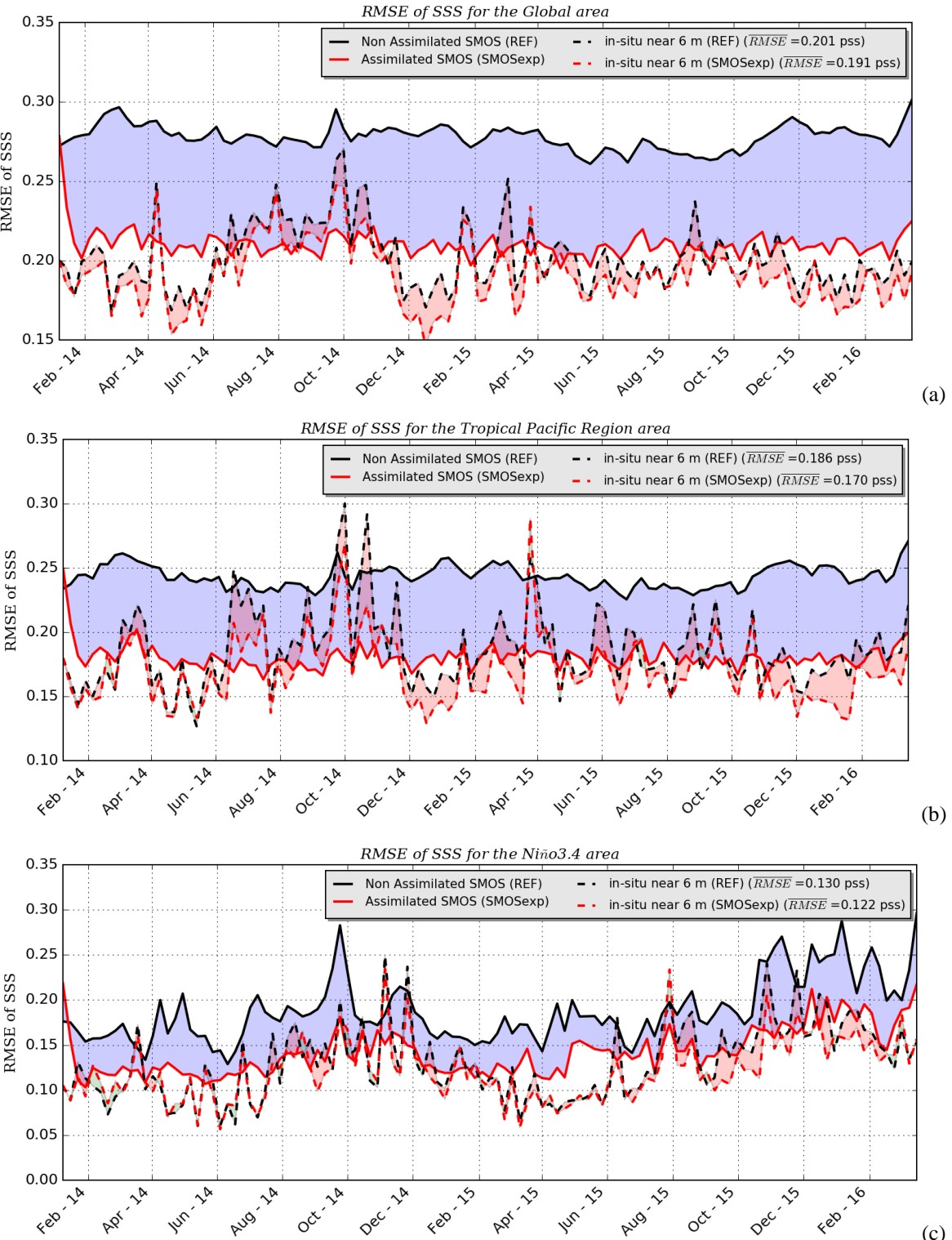

(a)

(b)

(c)

**Figure 6: RMSE of SSS with respect to SMOS data (solid lines) and RMSE of salinity near 6 meter depth with respect to in situ salinity data (dashed lines), in the 1-6 day forecast fields in REF (black lines) and SMOSexp (red line) in the global domain (a), the Tropical Pacific (b) and in the Niño3.4 region (c). RMSEs are evaluated for each weak and the mean $\overline{RMSE}$ of the in-situ salinity are denoted in the legend. The regions used here have south-west and north-east corners defined as: Tropical Pacific [30°S, 120°E] to [30°N,70°W]; Niño3.4 [5°S, 170°W] to [5°N, 120°W].**

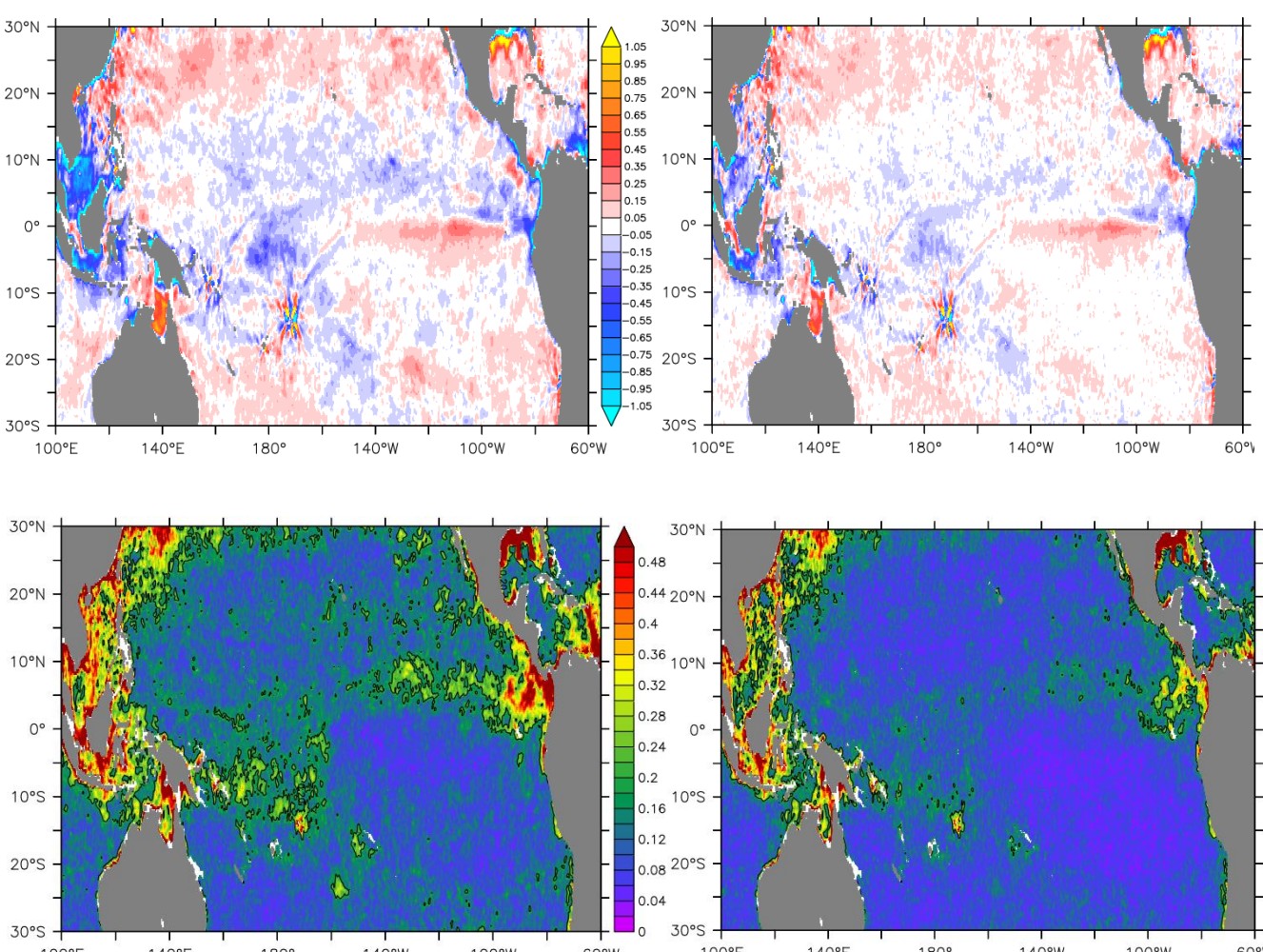

**Figure 7: Mean difference (top) and root-mean-square-difference (bottom) of monthly mean SSS (pss) with respect to the SMOS data (model minus SMOS) in the analysis fields in REF (left) and SMOSexp (right) experiments on 2015 year.**

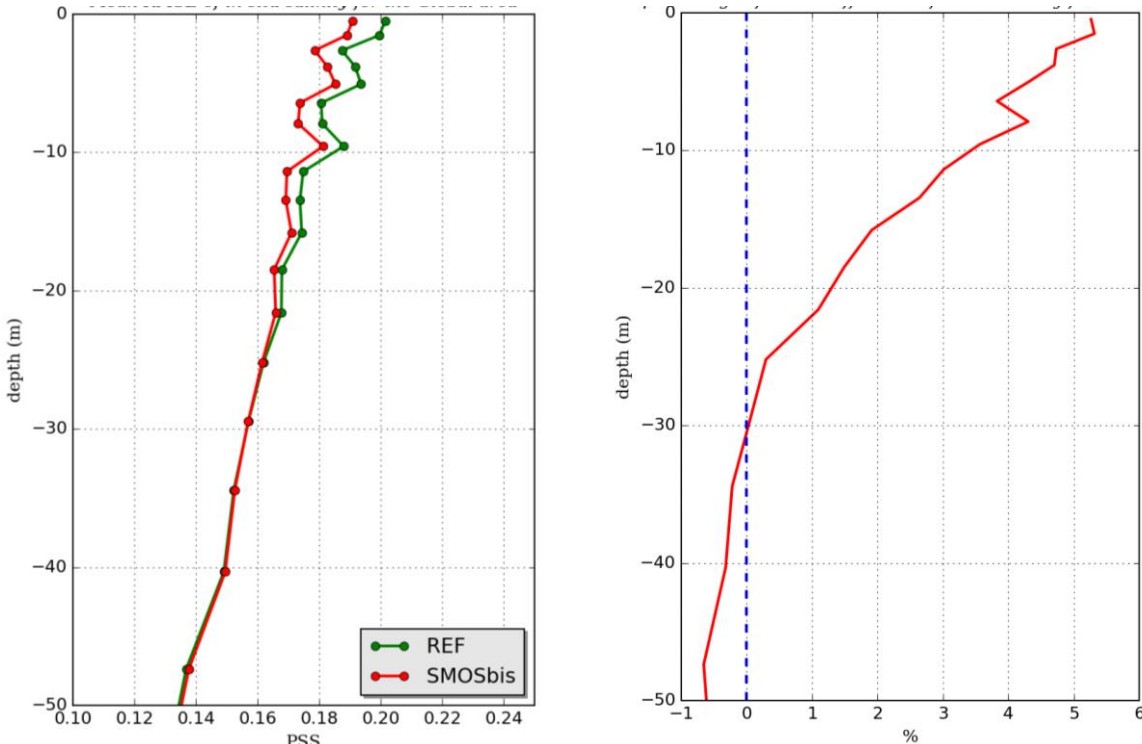

**Figure 8: Average salinity RMSE (pss) compared to all in situ measurements (left) over the period 1[st] Jan 2014 to 2[st] Mar 2016 in global domain for the *REF* (green line) and *SMOSexp* (red line) experiments as a function of depth over the top 50 m. The corresponding percentage of RMSE difference of all in situ salinity measurements between REF and SMOSexp experiments (right) (positive difference implies a reduction in RMSE by the SSS assimilation).**

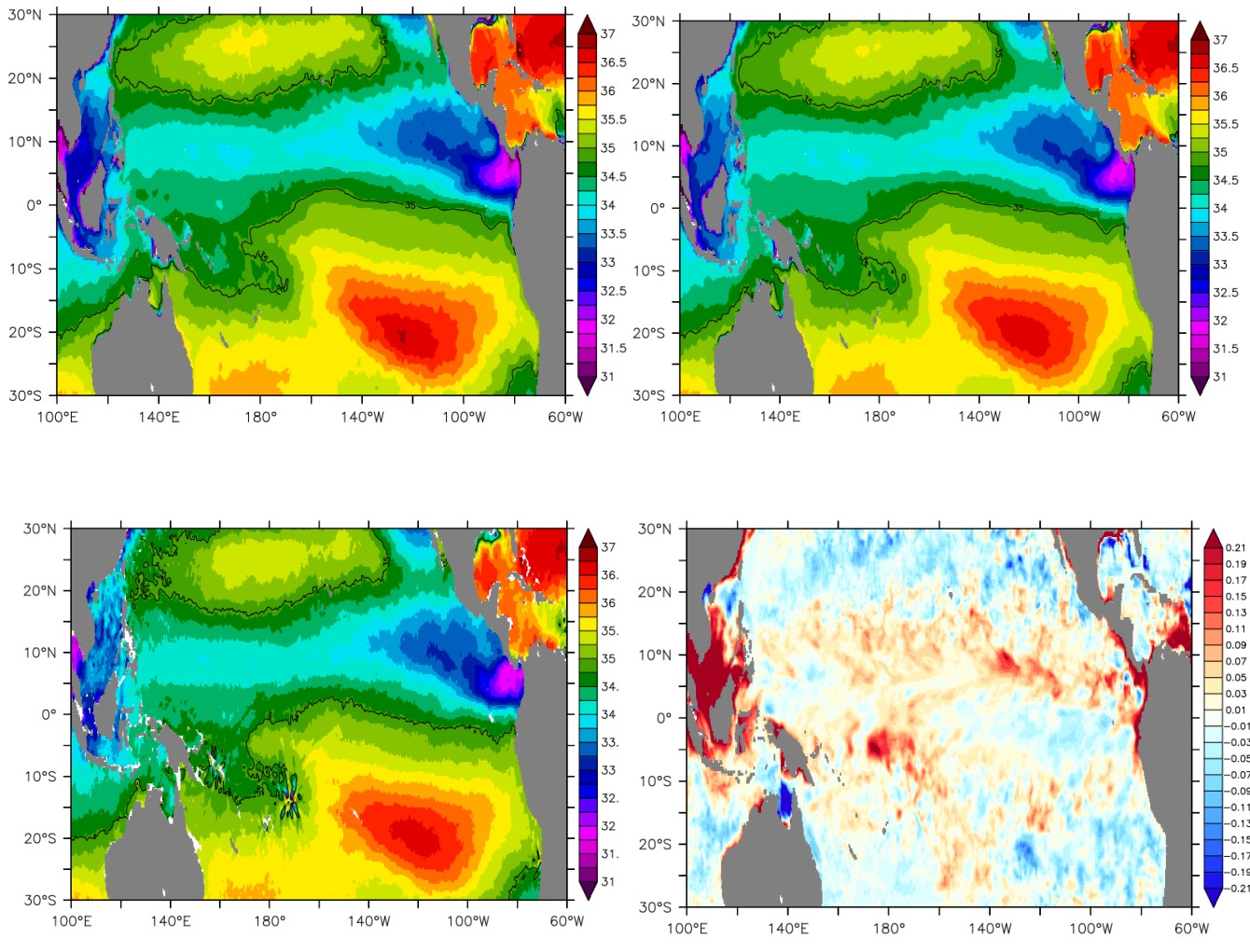

**Figure 9: Mean October 2015 SSS estimation from the REF experiment (top, left), the SMOSexp experiment (top, right), the SMOS SSS measurements (left, bottom) and annual mean difference (2015) between the SMOSexp and REF experiment (bottom, right). The isohaline 34.8 pss is the (black solid line) is represented.**

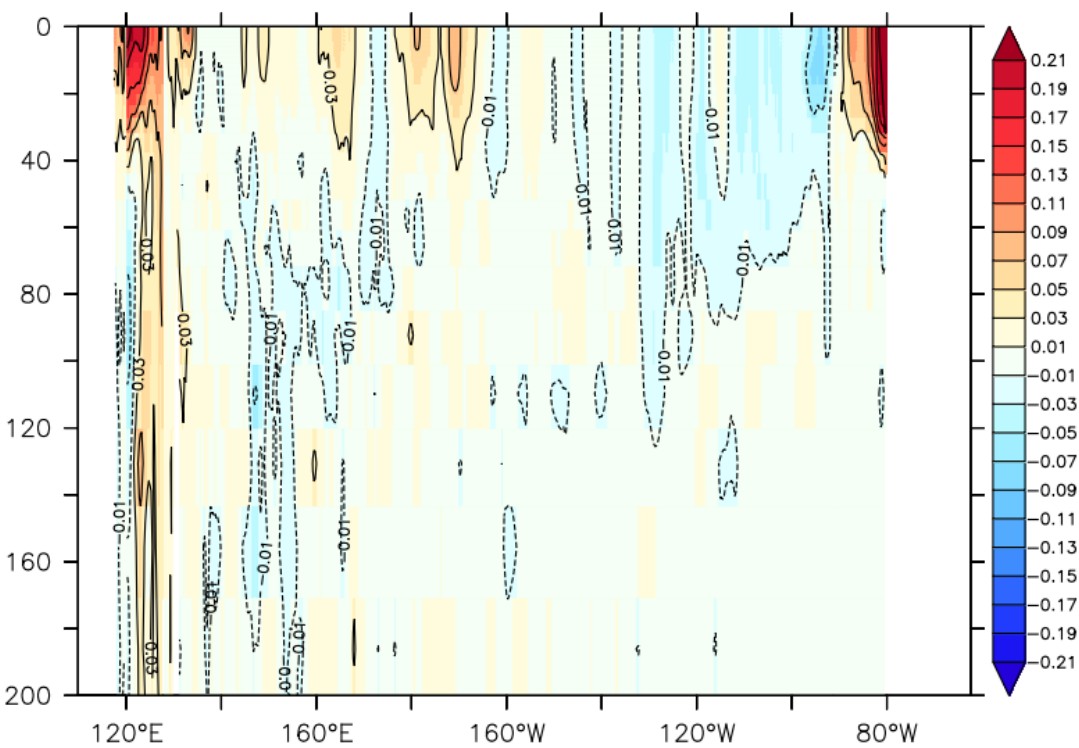

**Figure 10: Vertical section along the equator of the mean model salinity difference between the SMOSexp and REF experiments for the year 2015.**

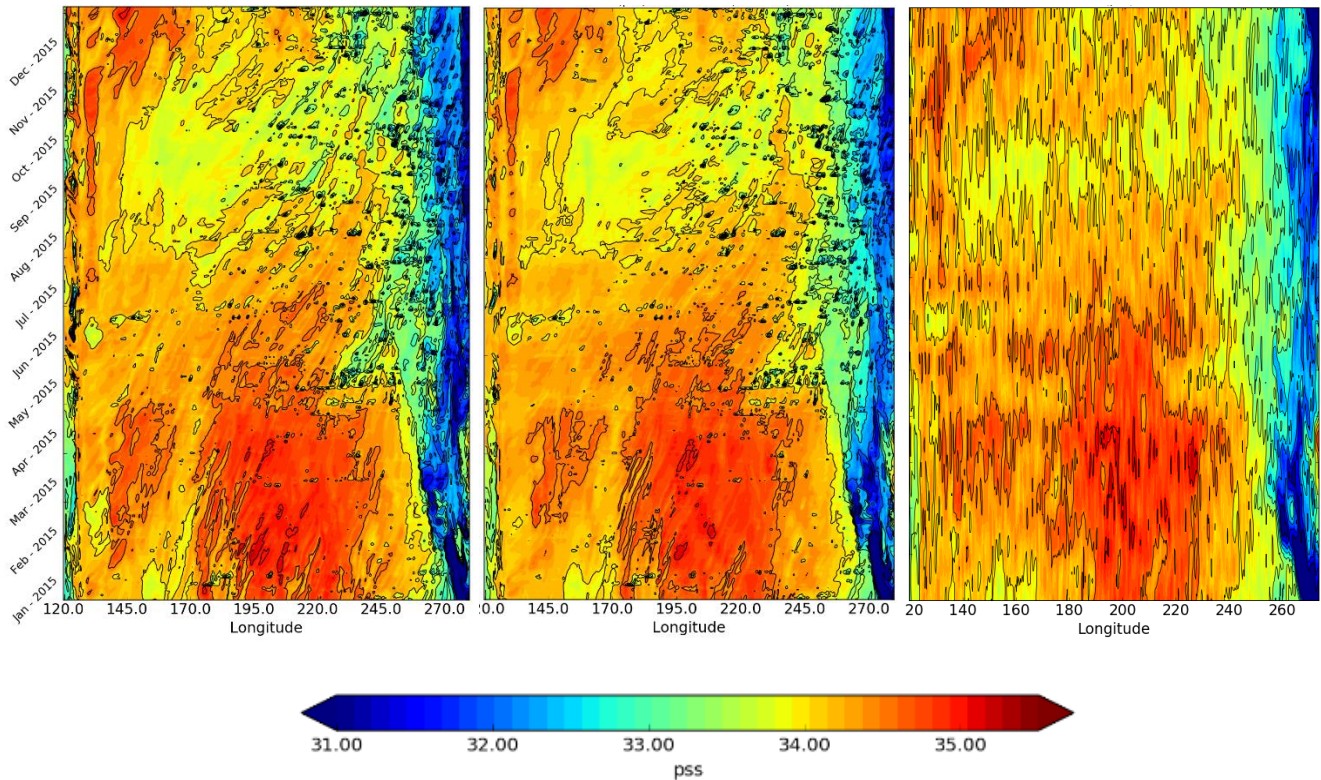

**Figure 11: Hovmöller of SSS at 5°N for the REF (left) and SMOSexp (middle) and SMOS data (right)**

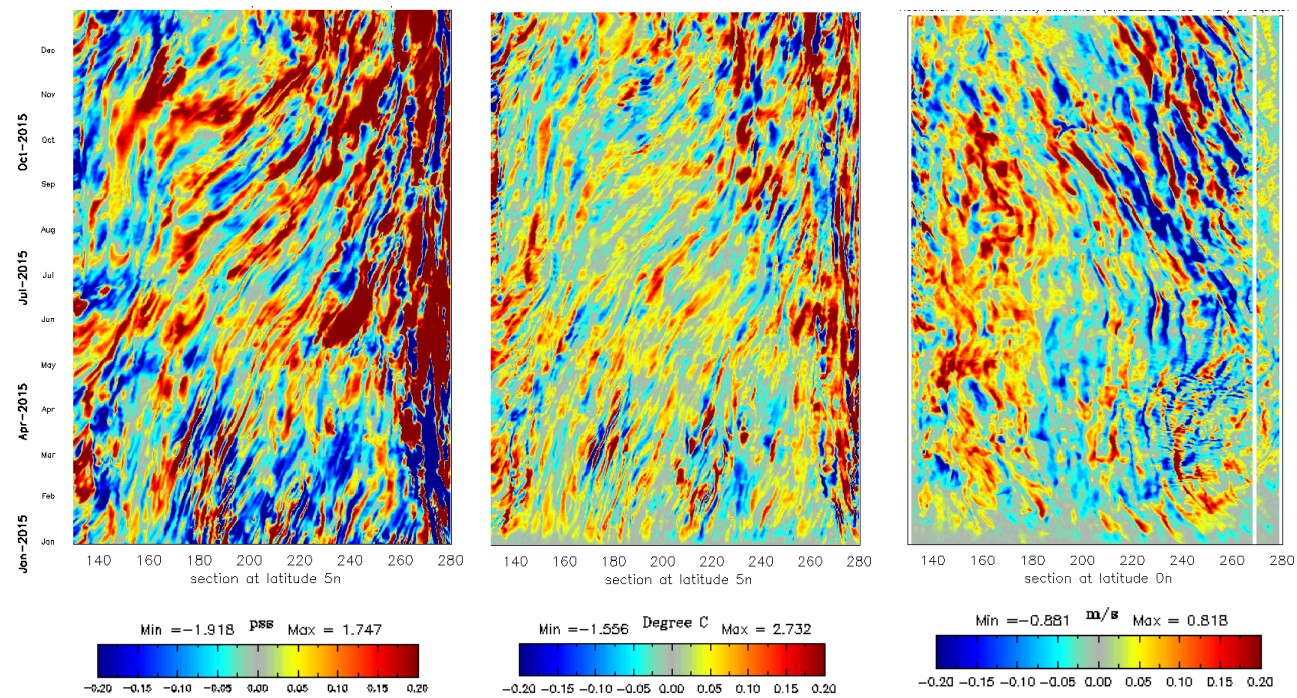

**Figure 12: Hovmöller ofdifferences in SSS (left), SST (middle) at 5°N and sea surface zonal velocity (U) (right) at the equator between the SMOSexp and the REF experiment in 2015.**

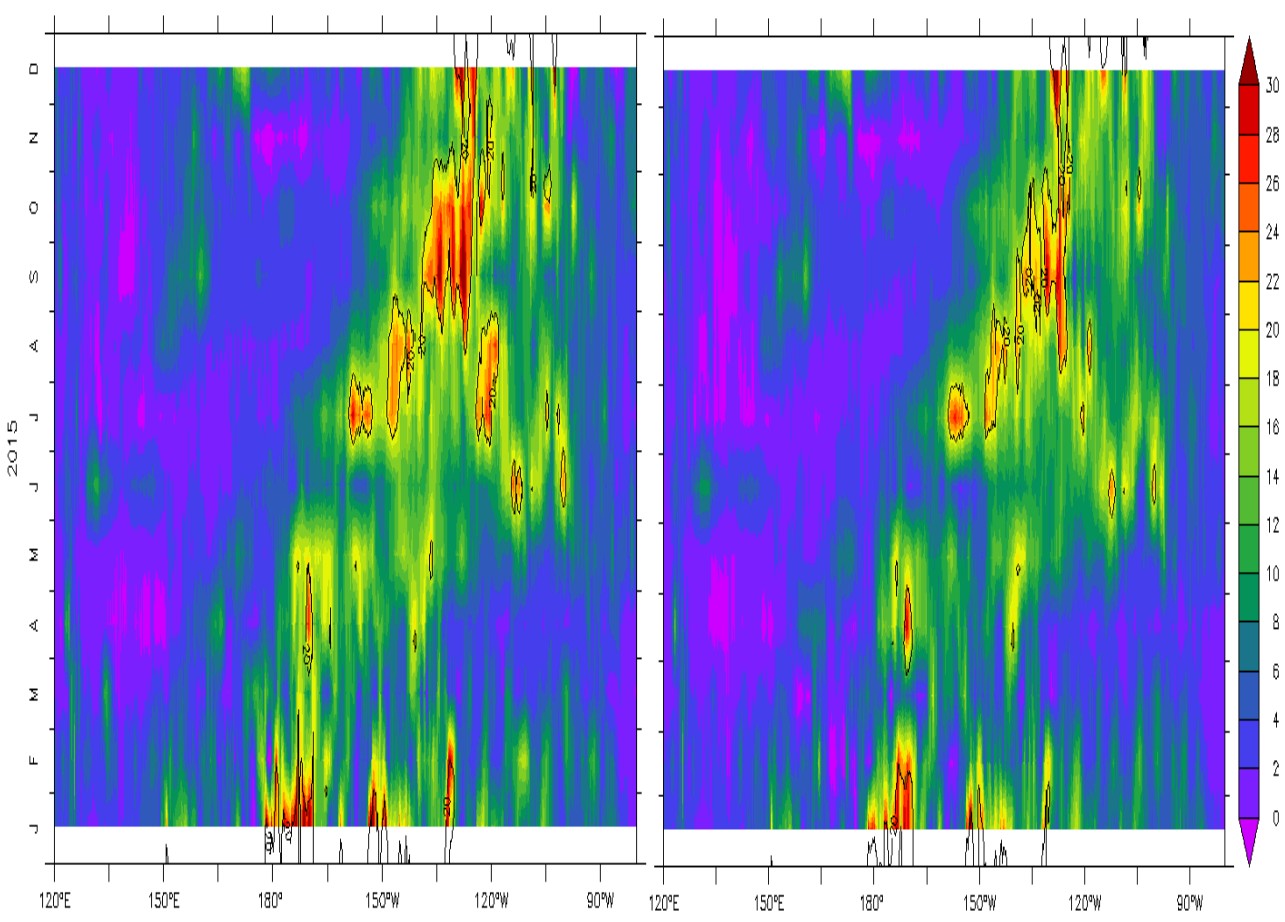

**Figure 13**: Hovmöller of Barrier Layer Thickness (BLT) at 5°N for the REF experiment in (left) and for the SMOSexp (right) experiment in 2015.

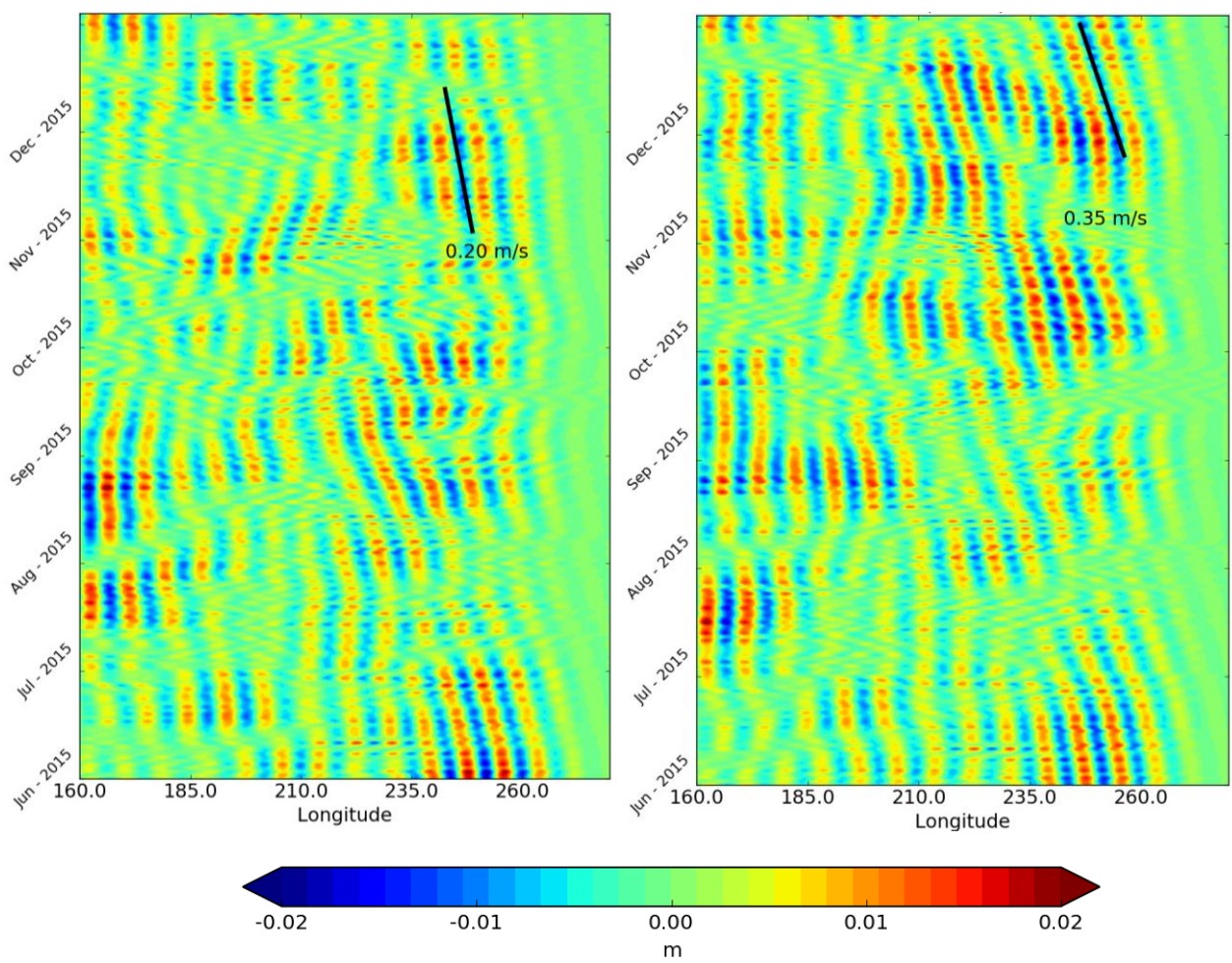

**Figure 14: Hovmöller of 28–40 day (33 days) band-passed SSH anomalies at 4°N referenced to the temporal annual mean of June-December 2015 for REF experiment (left) and for SMOSexp experiment (right). The propagation speeds of 0.20 and 0.35 m/s (solid lines) are representative of the propagation speed for the 28–40 day bands.**

.

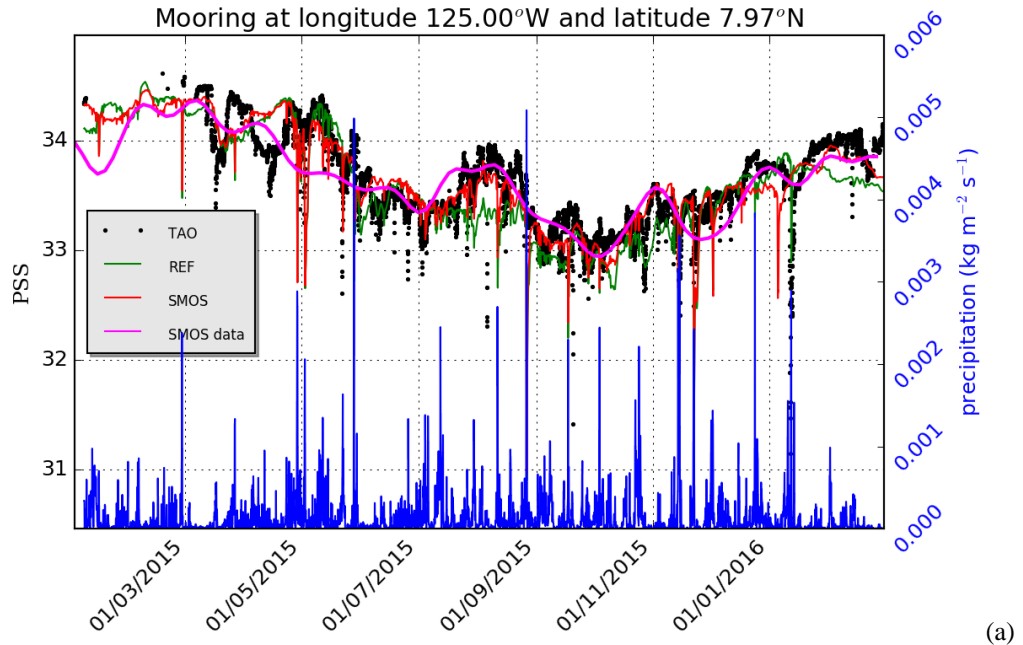

(a)

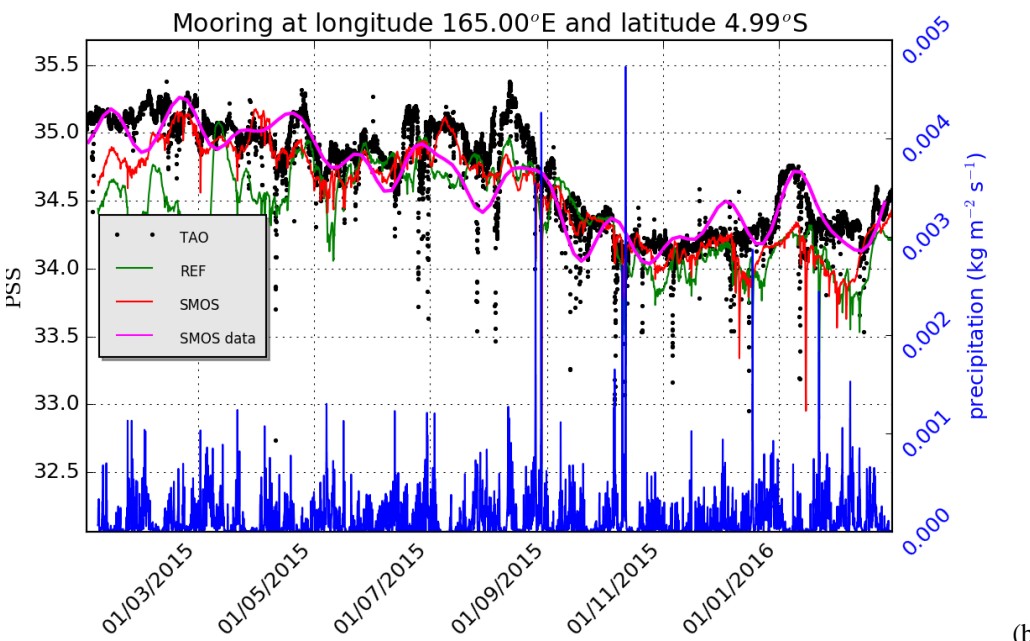

(b)

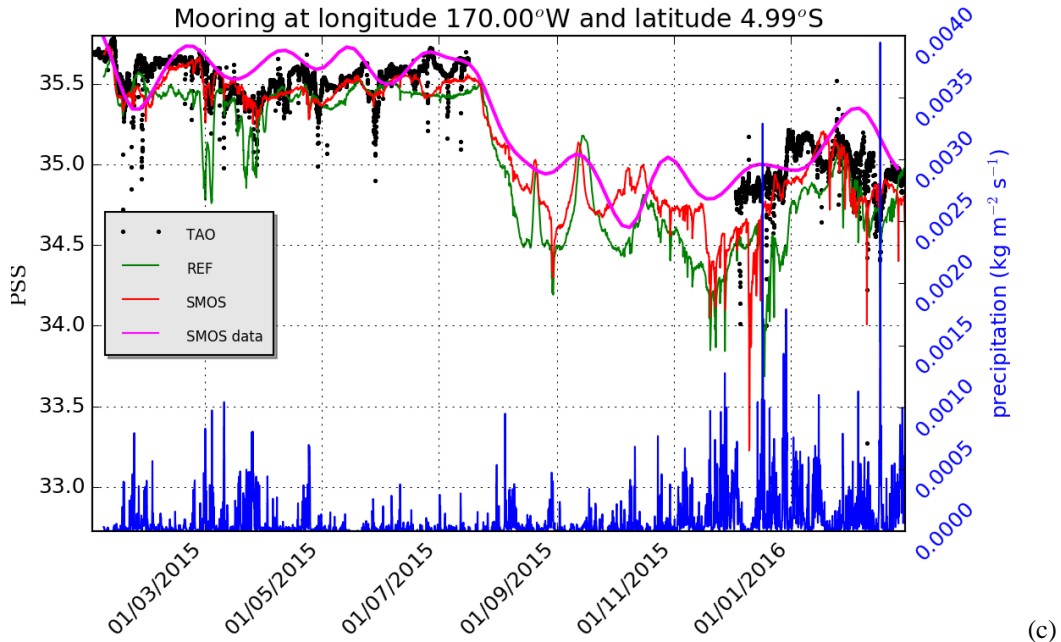

**Figure 15: Time evolution of the hourly TAO observed salinity (black), the hourly model REF (green), SMOSexp (red) simulations and the assimilated SMOS data (magenta) at three different TAO moorings locations, cold tongue (a) (125°W,7.97°N), warm pool (b) (165°E,4.99°S) and (c) salt front (170°W,4.99°S) from January 2015 to March 2016. The precipitation rate (blue line) coming from the atmospheric ECMWF forcing is superimposed**

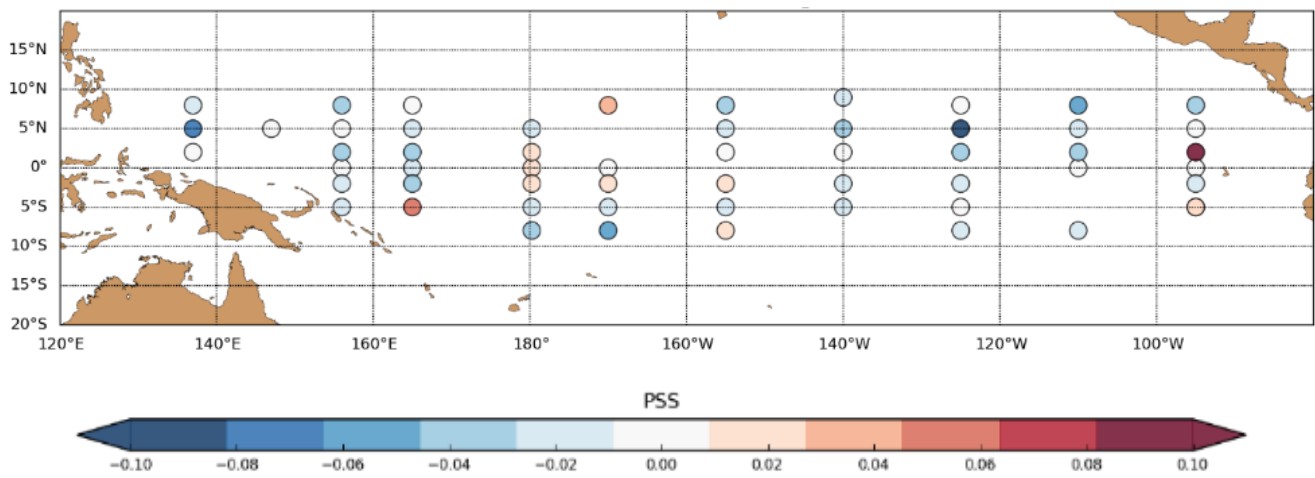

**Figure 16. Difference in model salinity RMSE (pss) at 1 m depth calculated against the 1 m depth TAO mooring salinity values (SMOSexp - REF) calculated over the period 1st Jan 2014 to 16th March 2016 (negative/positive difference implies a reduction/increase in RMSE by the SMOS assimilation). Moorings are only included if they have more than 1 week of measurements during the period.**

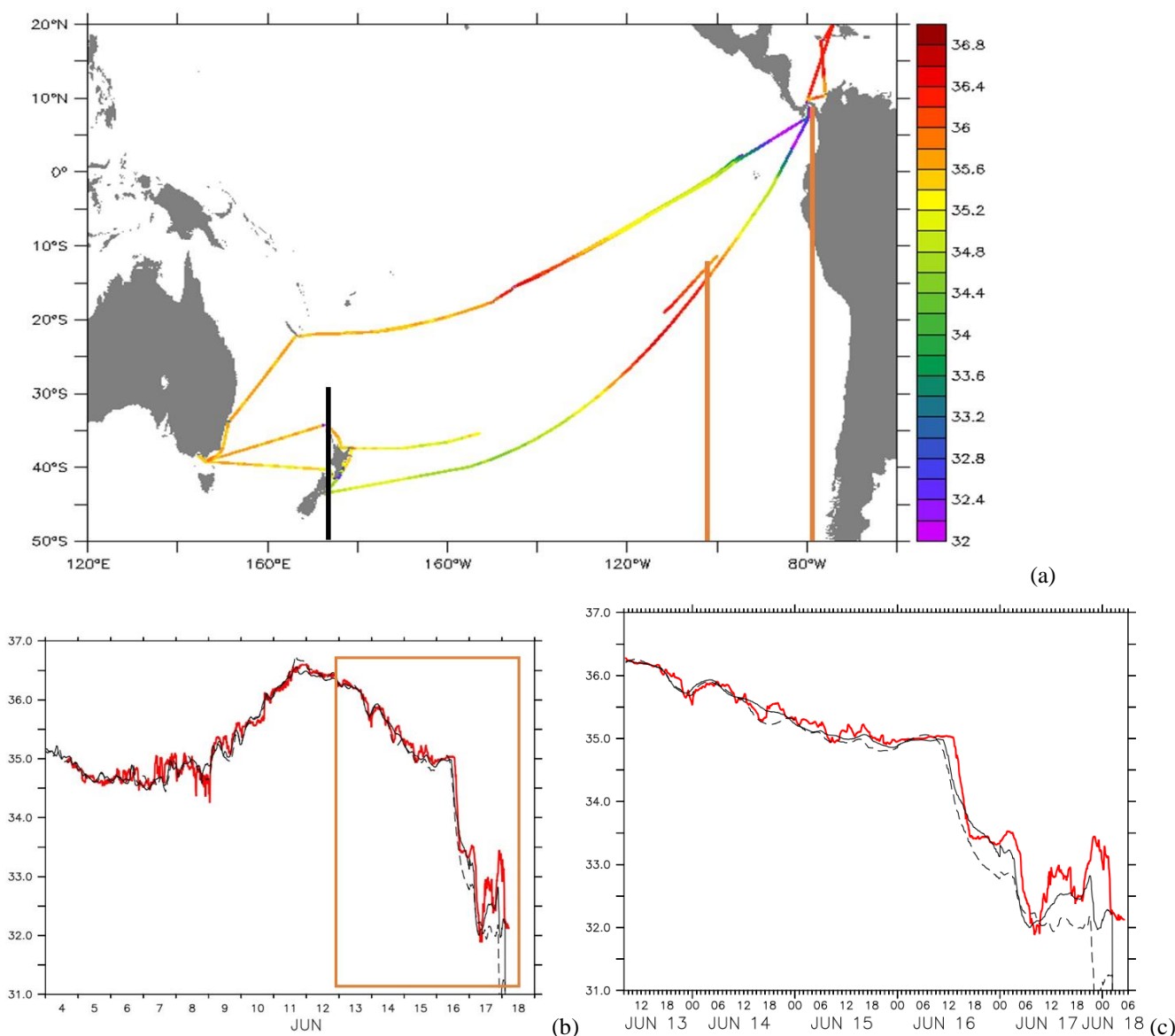

**Figure 17: Ship route of the Matisse with TSG salinity observations (PSS) (a) and TSG Salinity observations compared to near sea surface salinity analysis (b,c) from the OSEs (red line= observations, dashed line= REF, black solid line = SMOSexp). A zoom from the orange rectangle of (b) is shown in (c).**

| Instrumental errors ($R_{inst.}$) | |
|---|---|
| Altimetry | |
| JASON2, ALTIKA/SARAL | 2 cm |
| HAIYANG-2A | 4 cm |
| SST | |
| OSTIA L4 | 0.5°C |
| In-situ at sea surface | |
| XBT, moorings, Argo floats, sea mammals | 0.03°C and 0.0075 pss |

**Table 1:  Instrumental errors used for the current operational system.**

| Experiment name | Period | Assimilated observations | SSS product |
|---|---|---|---|
| Reference (REF) or control run | Jan 2014- March 2016 | Regular observation data without satellite SSS. | No SSS assimilation |
| SMOSexp | Jan 2014 - March 2016 | Regular observation data plus SMOS satellite SSS observations. | 4-day 0.25°x0.25° SMOS data from LOCEAN (L3-Debiased-Locean-v2) |

**Table 2: Experiment descriptions.**

| Regions [south-west to north-east corners] | Percentage of RMSE difference of SSS when SMOS SSS is assimilated and mean number of observations | | | |
|---|---|---|---|---|
| | SMOS SSS | | In situ salinity near 6 m depth | |
| | % | Mean number of obs./week | % | Mean number of obs./week |
| Global ocean | 24 % | 372,000 | 4.7 % | 1500 |
| Tropical Pacific [30° S, 120° E] to [30° N,70° W] | 26 % | 165,000 | 7.9 % | 500 |
| Niño 3.4 [5°S, 170°W] to [5°N,120°W] | 23 % | 9,500 | 4.8 % | 36 |
| Niño 4 [5°S, 160°E] to [5°N,150°W] | 22 % | 9,500 | 6.7 % | 38 |
| Niño 3 [5°S, 150°W] to [5°N,90°W] | 25 % | 11,400 | 3.3 % | 57 |
| North Tropical Pacific [8°N, 160°E] to [20°N,100°W] | 30 % | 22,300 | 10 % | 33 |
| South Tropical Pacific [20°S, 160°E] to [8°S,90°W] | 24 % | 24,000 | 6.6 % | 64 |

**Table 3: Percentage of RMSE difference of SSS for SMOS and for in-situ salinity at 6 m depth in different regions. The average number of SSS data assimilated per week is also indicated.**