# Peer review of "Data assimilation of SMOS observations into the Mercator Ocean operational system: focus on the Niño 2015 event"

_Ocean Science, 2018_

## Referee Comment (RC1) · Anonymous Referee #1 · 16 Nov 2018

Review of "Data assimilation of SMOS observations into the Mercator Ocean operational system: focus on the Nino 2015 event" by B. Benoit et al. (os-2018-113).

This manuscript describes the scheme of assimilating satellite SSS data in the operational ocean data assimilation system in Mercator Ocean, in particular, the scheme of correcting the bias of the model and satellite SSS. Then, they performed Observing System experiments (OSE), and demonstrate that the misfit between the model and in situ salinity data. They also reported that the equatorial surface zonal currents are affected by the SSS assimilation. The description on the scheme of SSS data assimilation is very comprehensive and valuable for the readers who are developing ocean data assimilation systems or exploit ocean data assimilation results. The OSE results indicate clear benefit of assimilating the satellite SSS data. However, the manuscript has severe problems on its presentation. So, I would like to recommend to perform a substantial revision according to the following comments. Then I expect that I still need to make several comments after the revision. I would like to withhold the judgement whether this manuscript is acceptable or not until it revised well.

**General comments**

1. The largest problem of this manuscript is English. Although I am not a native speaker, I do not think there are many grammatical mistakes or sentences which are not understandable. I suggest some modifications in the specific comments although I gave up to check the entire manuscript. I recommend authors to ask an English check preferably by native speakers.
2. I found this manuscript is not according to the general rule of writing academic paper in many places. For example, citations are often not according to the general rule. Abbreviations are not defined. They should check the general rule for writing and carefully modify the manuscript.
3. The section titles are often appropriate, and figure captions are often not understandable, and they do not include required information on the figure. They should carefully reconsider those things.
4. The abstract is not appropriate. It describes main result of the OSE results only very briefly and it does not address on the bias correction scheme of SSS. They should revise the abstract.
5. The introduction section also has bad structure. Things which are not directly

related to this paper is written in a substantial length, which should be shortened. The order of descriptions are not reasonable. I made some comments on the structure of the section in the specific comments. I recommend authors to revise the section according to those comments.

**Specific Comments**

1. P1, L9: I suggest to replace "most often closer to 5 meters deep than the surface" by "mostly closer to 5 meters depth than the surface"

2. P1, l12-14: I think the point addressed in this paragraph is not main pint of this paper and should be omitted. I think the explanation on SMOS-Nino 2015 project can also be omitted. Instead, authors can add the description on the bias correction scheme, and more detailed impacts of SSS data assimilation presented in this paper.

3. Original words for abbreviations should be specified in the main text even if they were specified in the abstract. Therefore, authors should specify the original words of SSS, SMOS, SMAP, etc.

4. P1, L25: What SMOS, Aquarius, and SMAP indicate should be described.

5. P1, L27: Authors should cite some references on the Mercator Ocean system.

6. P1, L27: I suggest to put "evaluated in the SMOS Nino 2015 project ((https://www.godae-25 oceanview.org/projects/smos-Niño15)" after "the Mercator Ocean Operational System", and remove the html address from P2, L25.

7. P1, L28: I suggest to replace "by the satellite SSS data assimilation, considering all of other ocean observing components" by "by assimilating the satellite SSS in addition to the observation data assimilated regularly in the operation".

8. P1, L29: I suggest to replace "The focus has been primarily on the 2015- 2016 El Niño event in the Tropical Pacific associated with strong SSS anomalies, seen in in both model and observations" by "The focus has been primarily on the 2015- 2016 El Niño event, in which strong SSS anomalies, are seen in the tropical Pacific in both model simulations and observations."

9. P2, L3: Authors missed to put tilde on the "n" of "El-Nino". (There are many mistakes the same as this.)

10. P2, L4: The term of NINO3 or NINO3.4 or NINO4 is generally used for the index indicating strength of ENSO based on SST anomaly instead of "ENSO index". In addition, definition of indices must be described or references must be cited.

11. P2, L5: Some citations are required for the Multivariate ENSO Index.

12. P2, L6: I do not know what authors means by "its onset in 2014 is visible." I do not understand what state the "visible" indicates.

13. P2, L6: I suggest to replace "It was more a Modoki El Nino (Ashok and Yamagata, 2009) than a « classical » one." to "It was more likely to be a Modoki El Niño (Ashok and Yamagata, 2009) or a central Pacific El Niño (Kao and Yu, 2009) than a classical eastern pacific El Niño."

14. P2, L9: Please add " in the year" after "but did not lead to development of an El Nino"

15. P2, L10: "As shown in (Corbett et al., 2017)" must be "As shown in Corbett et al.,(2017)". Authors do not make citations not in the general way or the general format in many places. They should check the rule of citations in this journal.

16. P2, L8-L24: The details of the process of development of the El Nino and of the difference between westerly wind bursts in 2014 and 2015 are not so related to the following contexts of this manuscript. Thus this paragraph should be shorten substantially.

17. P2, L23: "Note that recently, significant freshening was also observed around 20°N, (Hasson et al., 2018)." is not related to the later context of this manuscript and thus it should be removed.

18. P2 L28: I suggest to replace "(OSE)" by "(OSEs)".

19. P2, L29: I suggest to replace "The OSE approach consists of comparing two identical assimilation experiments except that one data set, here the satellite SSS, is withheld from

the analysis in one of the experiments." to The OSE approach is conducted by comparing two assimilation experiments which are identical except that the satellite SSS data are withheld in the analysis in one of the experiments.".

20. P2, L32: I suggest to replace "This is a commonly agreed approach to evaluate observation networks within the GODAE OceanView community" to "Similar OSE approaches are generally used to evaluate observation networks in the ocean data assimilation community of GODAE OceanView"

21. P3, L1: I suggest to replace "because of the various and complex biases that affect them, see (Köhl et al, 2014)." by "because of the various complex biases (see, Köhl et al, 2014)"

22. P3, L3: It is not clear what "data and methods" indicates. Does it mean "the observation instruments and data processing methods"?

23. P3, L9: BoB should not be defined if the abbreviation is not used later.

24. P3, L9: I think "still" should be removed.

25. P3, L12: It is better to put citations for "Careful analysis of the SSS data sets shows that a bias correction is needed before their assimilation."

26. P3, L16-28: Discussions in these paragraphs should be moved to the last paragraph of P2.

27. P3, L25: I suggest to replace "OSE experiments" by "OSEs".
28. P3, L25: I do not know which part of Lahoz et al. (2010) states on this point. It is better to cite the related chapter. And I need to know which pages indicates this point for reviewing. In addition, I know that the necessity of comparing with observations that are not assimilated for a fair assessment is pointed out in Fujii et al. (2015) Journal of Operational Oceanography.

29. P4, L2: I suggest to replace "OSE" by "OSEs".

30. P4, L4: I suggest to replace "we will describe in more detail the data assimilation components that were specifically developed or adapted for the SSS data assimilation." by

"we will describe the data assimilation components that were specifically developed or adapted for the SSS data assimilation in detail."

31. P4, L13, "a satellite-based large-scale correction of precipitation has been performed with climatological estimates from GPCPV2.1 rain-fall": This sentence is not clear. Does this mean that a bias correction based on satellite-based climatological estimates from GPCPV2.1 rain-fall is applied to the precipitation fields of the ECMWF system?

32. P4, L16: I think "coastal runoff and 100 major rivers" should be "coastal runoff from 100 major rivers".

33. P4, L18: Citations are required for the Community Land Model Version 3 (CLM3).

34. P4, L18: I suggest to replace "At high latitude" by "At high latitudes"

35. P4, L24: I suggest to replace "Current Network" to "Regular observation data" or "Regularly assimilated observation data".

36. P4, L27: "L4" should be explained as well as "L2" and "L3".

37. P5, L28: I do not know what "progressively" means.

38. Caption of Table 1: I suggest to replace "current operational network" to "current operational system".

39. I recommend to merge Figures 2 and 3.
40. P6, L3: The method how to specify the representativity errors is not described.

41. P6, L9: "minimum" should be "minimizer".

42. P6, L9: Equation 1 should be put right after "the cost function given by". And explanation of Equation 1 should be re-structured in a more comprehensive way.

43. P6, L10: I suggest to replace by "(it is the mean innovation over 1 month on a 1°x1° grid 10 between 0 and 10 meters depth and the mean is symbolized by <>)" by "(it is the mean innovation over 1 month in a 1°x1° box between 0 and 10 meters depth)". The symbol

of mean should be explained later.

44. Equation (1b): I think this equation is not correct. I think "H$\boldsymbol{\xi}$" in the second term may be replaced by "Hx".

45. P6, L24: "The Fig.4" should be "Figure 5".

46. I recommend to merge Figures 4 and 5.

47. P6, L25: I think it is natural that the SSS bias and the model bias is not similar. I think it would be rather strange if these two bias fields were similar.

48. P6, L26: "SMOSexp" is not defined yet.

49. P7, L7-9: I do not understand the explanation of the iterative boot-strap method.

50. P7, L10: I am not sure whether the increase over the regions with sparse in-situ data is originally included in $\boldsymbol{R}_\xi{}^o$.

51. Fig. 7 is not referred in the text.

52. P7, L20: I suggest to replace "OSEs experiment design" to "OSE design" Since OSE includes "experiment". The same mistake is found at P8, L1.

53. P8, L5: "TOA/TRITN" should be "TAO/TRITON".

54. P8, L8: I suggest to replace "Evaluation of the analysis toward assimilated observations" by "Evaluation of the reduction of the misfit from the assimilated data in the analysis".

55. P8, L10: I suggest to replace "by the SSS data set assimilated" by "by assimilating satellite SSS data".

56. P8, L11: "between" should be replaced by "from".

57. Figure 8: I suggest to modify the figure caption as follows: "RMSE of SSS with respect

to SMOS data (solid lines) and RMSE of salinity at 5 meters depth with respect to in situ salinity data (dashed lines), in the 1-6-day forecast fields in REF (black lines) and SMOSexp (red lines), in the global domain (top), the Tropical Pacific (middle), and in the NINO3.4 region (bottom). RMSEs are evaluated for each weak and the averaged counts of the observations used for the calculation of RMSEs are denoted in the legend." In addition, definitions of "Tropical Pacific" and "NINO3.4 region" must be addressed.

58. Figure 9: I suggest to modify the figure caption as follows: "Mean difference (top) and root-mean-square-difference (bottom) of daily mean SSS with respect to the SMOS data (model minus SMOS) in the analysis fields in REF (left) and SMOSexp (right) on 2015 year."

59. P8, L26: I suggest to replace "Fig. 9 shows the mean and standard deviation of the daily ?? or monthly differences between the (analyzed) SSS for REF and SMOSexp simulations compared to the SMOS SSS observations (non-debiased)" by "Fig. 9 shows the mean and root-mean-square differences of daily mean SSS in the analysis fields in REF and SMOSexp compared to the non-debiased SMOS data."

60. P8, L29: I suggest to shorten this paragraph as follows: "The mean SSS bias in REF exhibits large scale patterns, coinciding with the 2015 SSS anomaly for the open ocean (Fig. 1). Large bias is also found in the Indonesian Archipelago. In contrast, the bias is effectively reduced in SMOSexp as well as the root-mean-square differences."

61. P10, L5: It would be better if what induces the SST rise and why the migration of the warm water pool is accelerated are discussed. The impact on the barrier layer can be presented with the figure on the zonal currents.

62. Figure 15: It is difficult confirm the faster propagation of TIWs by comparing the 2 panels at the right. And the left panel are not discussed and should be omitted.

63. Figure 16: Positions of the TAO/TRITON buoys should be clarified.

64. P11, L12: "West (East) Tropical Pacific" should be "western (eastern) tropical Pacific".

65. P11, L18: I suggest to modify "SSS-OS) (http://www.legos.obs-mip.fr/observations/sss)" to "(SSS-OS; http://www.legos.obs-mip.fr/observations/sss)".

66. Figure 18: The Matisse ship route should be indicated in this figure. And I recommend to merge Figs. 18 and 19.

---

## Referee Comment (RC2) · Anonymous Referee #2 · 12 Dec 2018

The paper discusses the impact of the assimilation of SMOS data on the Mercator Ocean model.

The uptake of SMOS data in the scientific community and the ocean modelling community is relatively slow due to many unique challenges associated to remote sensed salinity observations.

Therefore such paper comparing the model results with and without assimilation of SMOS data is quite important and timely. The presented study is well planned and executed and followed by an independent validation.

My major critique points are the following 2:

[Figure]

A. The description of the data assimilation scheme is rather confusing, partially due to the fact that two different schemes are used together (3D-var for the bias and the reduced order Kalman Filter). Having two clearly distinct sections (one KF section and one 3D-Var) and for each of these section the relevant subsection (model error covariance, observational error covariance,...) would already clarify a lot.

B. The interpretation of the results is a rather superficial. The discussion on why the model has a salinity bias relative to in situ observations should be substantially improved. Also did the authors check of a possible degradation in other model variables (besides salinity) due to the assimilation of SMOS data?

In addition, the overall quality of the figures should be significantly improved. The font-size is really small and the text is barely readable on a print-out (especially on Figure 18). As a rule of thumb, the font-size of the figure after embedding it in the document should be roughly the same font size as the text. Also the text labels should not overlap with other text (Figure 18). Labels on figure 15 are blurry and also very hard to read (probably JPEG compression artifacts; vector image formats like EPS or PDF provide a much better quality).

Therefore I recommend major revision before publication in Ocean Science.

Specific comments:

page 5, line 27: "The localization of the error covariance is performed assuming a zero-covariance beyond a distance defined as twice the local spatial correlation scale"

How is the local spatial correlation scale determined? Is it derived from the ensemble and does it vary then for every model grid point?

page 6, line 9: "...where x is the in-situ bias to estimate, B is the background error covariance of the 3D bias, d is the innovation vector (it is the mean innovation over 1 month on a $1° \times 1°$ grid between 0 and 10 meters depth and the mean is symbolized by <>)"

Why are only near-surface in situ values used? If the in situ data are assumed to be biased at the surface, should one not also expect a bias at depth?

Figure 4, panel b: What is the negative anomaly at 20°S and 120°W?

section 2.3.4 "SSS observation error". It seems that the same SSS observation error is used in the 3D-var scheme and the reduced-order KF despite the fact they use different background-error covariance and different time scales. Should the representativity error not be quite different if one uses different time-scales (bias versus instantaneous field)?

page 10, line 10: "A reinforcement of the TIWs (the slope is steeper)..." Do you mean a reinforcement (related to amplitude and energy) or acceleration (related to speed)? Also in a Hovmöller-Diagram, one plots space in the x-axis and time in the y-axis. The slope is thus the *inverse* of the speed. Figure 15 seems indeed to show that the slope get less steep in the second half of 2015. Please provide a discussion about why we see a change in the propagation speed and quantify the changes that you are seeing.

page 11 and Figure 17: The impact of the SMOS SSS assimilation seems to depend on the latitude. The impact at latitudes lower than 5°N (or 5°S) seem to be consistently higher than near the equator. Why?

Minor comments:

page 6, line 14: "Spatial correlations in B are modeled by means of an anisotropic Gaussian recursive filter." Please provide the equations on how this filter is built.

page 6, line 18: in the definition of d_\xi why is \xi transposed?

page 6, line 25: "The Fig. 4 shows the SSS bias ($\xi$ term of the Eq. 1). The patterns are different than the model bias (Fig. 4) and often of opposite sign but have the same magnitude..." Should that not be "Fig. 5" showing the SSS bias?

Figure 4: Please add a color-bar on Figure 4.

General comment about citation: "Following (Desroziers et al., 2005),..." should be "Following Desroziers et al. (2005),...". Please correct also other similar citation issues.

Salinity is sometimes expressed in PSS (practical salinity scale), for example page 8, line 18 and sometimes in psu (practical salinity unit), for example page 9, line 16. Please make use of the same unit.

page 8, line 26: "Fig. 9 shows the mean and standard deviation of the daily ?? [sic] or monthly differences between the (analyzed) SSS for REF and SMOSexp simulations compared to the SMOS SSS observations (non-debiased). Statistics are computed over the year 2015 for the Tropical Pacific Ocean." Daily or monthly?

page 10, line 6: "At the equator, there is an acceleration of the Warm Water Pool migration towards the east (Fig.14c) which helps the ocean-atmosphere coupling and thus the triggering of El Niño.". This is not so clear to see from the figure. Can you give the start and end latitude of the Warm Water Pool migration?

Hoevmuller -> Hovmöller (or Hovmoeller)

define acronym TIW

---

## Author Comment (AC2) · 30 Jan 2019

See the pdf file that contains answers to the reviewer and the revised manuscript.

Please also note the supplement to this comment:
https://www.ocean-sci-discuss.net/os-2018-113/os-2018-113-AC2-supplement.pdf

---

## Author Response (AR1)

We would like to thank the anonymous reviewer#1 for the comments and suggestions, which have helped improve the clarity and readability of this paper notably.

**ANSWERS TO REVIEWER# 1**

**General comments:**

1. Some modifications and an english checking have been made. See changes.
2. General rules have been applied. See changes in all the document.
3. Some captions have been changed in order to better describe the figures.
4. The abstract has been rewritten in order to describe main results and also the bias correction issues have been included.
5. The introduction has been shortened and rearranged as suggested by the reviewer according to the specific comments.

**Specific comments:**

**1.:** Corrrection made

**2.:** The abstract has completely changed, see changes.

**3-10:** Modifications have been made**.**

**11-12**: Following recommendations of 2 reviewers, the introduction has been changed and shortened**.**

**13-15:** Modifications have been made

**16:** This paragraph has been shortened**.**

**17:** This sentence has been removed.

**18-21:** Modifications have been made

**22:** The sentence has been changes by**:** "The signal to noise ratio is still not high today, thus retrievals algorithms must be improved."

**23-30:** Suggestions were taken into account.

**31:** The sentences have been modified**.** Due to large known biases in precipitation, a satellite-based large-scale correction of precipitation is applied to the precipitation fluxes. This correction has been inferred from the comparison between the Remote Sensing Systems (RSS) Passive Microwave

Water Cycle (PMWC) product (Hilburn, 2009) and the IFS ECMWF precipitation (Lellouche et al., 2013).

**32-36:** Suggestions were taken into account**.**

**37:** The term "progressively" has been changed by "incrementally " that refers to the incremental Analyse Update (IAU).

**38-39:** Modifications have been made

**40:** A sentence has been added to explain how the representativity error is inferred: "Representativity errors concerning in situ observations were calculated a posteriori from a reanalysis over the period 2008-2012. The estimated errors are constant throughout the year. The method (Desroziers et al., 2005) consists of the computation of a ratio, which is a function of observation errors, innovations and residuals."

**41.:**  change made.

**42.:** Equations and text have been changed to clarify the purpose.

**43.:** change made.

**44**: No, it is correct.  This term $\xi$ refers to the satellite SSS bias we want to minimize which is different from x that refers to the salinity bias from surface in-situ data.

**45-46**: Done.

**47.:** We tried to clarify the 2 biases: The salinity in-situ bias at surface (x) and SSS (SMOS) bias ($\xi$) are different.  The new figures 3 show the in-situ bias without the SSS (SMOS) bias (a), the SMOS SSS bias from space only (b) and the in-situ bias with the SSS SMOS bias.

**48.:** done

**49**: We have improved the explanation of the boot-strap method.

**50.:** As written, the a priori error $R_{\xi}^{o}$  is a combination of a zonally varying error, together with an increase over regions with sparse in-situ data and near the coast. The algorithm to increase the error in case of sparse data and near the coast is not detailed in this paper.

**51-60**:  Modifications have been made

**61-62**: This paragraph and Figures have changed.

**63-65:** Modifications have been made

**66:** The Matisse ship route location used for the comparison is shown. Figures and caption have been merged.

We would like to thank the reviewer #2 for the comments and suggestions, which have helped improve the clarity and readability of this paper notably.

**ANSWER TO REVIEWER# 2**

**Major Comments**

A: *The description of the data assimilation scheme is rather confusing, partially due to the fact that two different schemes are used together (3D-var for the bias and the reduced order Kalman Filter). Having two clearly distinct sections (one KF section and one 3D-Var) and for each of these section the relevant subsection (model error covariance, observational error covariance,...) would already clarify a lot.*

**Answer:** Sections have been changed for clarity. Now, two different sections exist, a KF section with information on errors and one section on the bias correction scheme with a specific sub-section on the SSS error estimation.

B. *The interpretation of the results is a rather superficial. The discussion on why the model has a salinity bias relative to in situ observations should be substantially improved. Also did the authors check of a possible degradation in other model variables (besides salinity) due to the assimilation of SMOS data?*

**Answer:** This comment has been taken into account. The discussion on salinity biases has been improved to clarify the different processes and outcomes. In this paper a focus is given on the SSS bias and salinity error improvement during El-Niño 2015. The impact on the other variables of the model is often shown to be neutral and mainly on the surface with regard to data assimilation diagnostics. On the other hand, as in previous studies, it is shown that an impact exists on the propagation of TIWs through the modification of surface currents and stratification.

*In addition, the overall quality of the figures should be significantly improved. The font size is really small and the text is barely readable on a print-out (especially on Figure 18). As a rule of thumb, the font-size of the figure after embedding it in the document should be roughly the same font size as the text. Also the text labels should not overlap with other text (Figure 18). Labels on figure 15 are blurry and also very hard to read (probably JPEG compression artifacts; vector image formats like EPS or PDF provide a much better quality).*

**Answer:** We tried to improve all the font size on the Figures. Figure 18 and 15 have been changed.

**Specific comments:**

*page 5, line 27: "The localization of the error covariance is performed assuming a zero-covariance beyond a distance defined as twice the local spatial correlation scale"*

*How is the local spatial correlation scale determined? Is it derived from the ensemble and does it vary then for every model grid point?*

**Answer:** As said in Lellouche et al., 2013 and more recently in Lellouche et al., 2018b, the correlation scales (longitude, latitude, time) are deduced a posteriori from the Mercator Ocean global ¼° reanalysis GLORYS2V1 (GLobal Ocean ReanalYsis and Simulation, Ferry et al., 2012). These scales are used to define an "influence bubble" around the analysis point in which data are also selected and consequently vary for every model grid point.

Reference: Ferry, N., Parent, L., Garric, G., Bricaud, C., Testut, C. E., Le Galloudec, O., Lellouche, J. M., Drevillon, M., Greiner, E., Barnier, B., Molines, J. M., Jourdain, N. C., Guinehut, S., Cabanes, C., and Zawadzki, L.: GLORYS2V1 global ocean reanalysis of the altimetric era (1992–2009) at meso scale, Mercator Newsletter 44, 29–39, 2012.

*page 6, line 9: "...where x is the in-situ bias to estimate, B is the background error covariance of the 3D bias, d is the innovation vector (it is the mean innovation over 1 month on a 1 ◦ x1 ◦ grid between 0 and 10 meters depth and the mean is symbolized by<>)"*

*Why are only near-surface in situ values used? If the in-situ data are assumed to be biased at the surface, should one not also expect a bias at depth?*

**Answer:** This section has been modified. Now, there is a section on the bias correction. The first part discusses the actual operational bias correction scheme applied to in-situ 3D T/S profiles. The second part discusses the

addition of the extra term denoted $\xi$ to take into account biases in the satellite SSS.

As mentioned in the in-situ T/S section, the salinity bias is a 3D bias, therefore a depth salinity bias can potentially induce a surface salinity bias, see the next answer.

*Figure 4, panel b: What is the negative anomaly at 20 ◦ S and 120 ◦ W?*

**Answer**: A persistent large innovation at 11m, 41 m and 79 m depths, see (Figure 1 below) may induce a larger bias of salinity (negative anomaly) at sea surface. This is the case with the example given in Figure 3b,3c where a surface bias is seen from a certainly improper in-situ profile at depth near 120°W/20°S. Note also that this negative anomaly disappears the week after, see Figure 2 below.

[Figure]

*Figure 1 : Monthly innovation of salinity at 11 m (top) depth and 41 m depth (bottom) used in the bias correction scheme on week 13-20 (left) and week 20-27 (right) of January 2016. Red circle indicated the persistent and large anomaly near 120°W/20°S*

[Figure]

*Figure 2 : Salinity bias of salinity at sea surface on 2016/02/03, i.e, one week after the Figure 3b (paper).*

*section 2.3.4 "SSS observation error". It seems that the same SSS observation error is used in the 3D-var scheme and the reduced-order KF despite the fact they use different background-error covariance and different time scales. Should the representativity error not be quite different if one uses different time-scales (bias versus instantaneous field)?*

**Answer:** It is true, but as said in the text, to get an optimal set of parameters (weights, spatial scales and errors), several estimations were performed. These tests have been done with and without the bias correction in order to check the estimated errors. An off-line analysis is systematically done after the data assimilation from the Desroziers ratio (Desroziers et al., 2005) and allows to verify our hypothesis, i.e., we use the estimated SSS error from the bias correction (3D-var) in the data assimilation scheme (SEEK filter).

*page 10, line 10: "A reinforcement of the TIWs (the slope is steeper)..." Do you mean a reinforcement (related to amplitude and energy) or acceleration (related to speed)?*

**Answer:** The Figure changed for clarity. From this new figure the reinforcement of the TIWs is shown and is related both to the increase of the amplitude and the propagation speed.

*Also in a Hovmöller-Diagram, one plots space in the x-axis and time in the y-axis. The slope is thus the \*inverse\* of the speed. Figure 15 seems indeed to show that the slope get less steep in the second half of 2015. Please provide a*

*discussion about why we see a change in the propagation speed and quantify the changes that you are seeing.*

**Answer:** Figures have been changed for clarity. The period only concerns the second half of the year and TIWs are seen to propagate in both cases with different magnitudes at different propagation speed. It is true that a steeper slope means a lower propagation speed. As shown in the Figure, the change is important in the last part of the year (October to December) in the Central Pacific where the magnitude and the propagation speed of TIWs decrease in the REF experiment. Note that, on a shorter period (August and September), the amplitude of TIWs decreases in SMOSexp.

The new Figure 12c shows the difference of the zonal current at the equator between 2 experiments. The positive pattern in the western Pacific in the first 10 months of the 2015 year shows that the eastward advection is reinforced. This is also enhanced by an Eastward propagation in the Eastern basin during Autumn 2015.

A vertical section of the mean zonal current (June-Dec. 2015) is shown for the 2 experiments at 210°W in Figure 3. The North Equatorial CounterCurrent NEEC (red) and South Equatorial Current (SEC) (particularly the branch north) (blue) are shown. As expected, a stronger NECC is seen and could block the propagation of TIWs. In addition, the SEC is a little stronger in SMOSexp than in REF hence the acceleration of TIWs.

[Figure]

*Figure 3 : Section of the mean Zonal Current at 210°W (0-100 m) for the REF (left) and SMOSexp (right) experiments between June and December 2015.*

The new Figure 13 (the time-space evolution of Barrier Layer Thickness at 5°N) shows also the link between the faster propagation of TIWs. The Eastern and Central Pacific are saltier which induces a decrease of the stratification, the

MLD increase that corresponds to a decreased BLT. This effect could induce an acceleration of the TIWs with a mixing enhancement.

*page 11 and Figure 17: The impact of the SMOS SSS assimilation seems to depend on the latitude. The impact at latitudes lower than 5 ◦ N (or 5 ◦ S) seem to be consistently higher than near the equator. Why?*

**Answer:** This Figure changed. The entire period is considered (2014-March 2016) and the color-bar is reversed: negative/positive difference implies a reduction/increase in RMSD by the SMOS assimilation. The remark is interesting and is linked to Figure 9d. Actually, the impact of SMOS SSS assimilation is larger in the ITCZ and SPCZ regions, it reflects the over estimation of E-P that the data assimilation tends to correct (SMOSexp is saltier in regions where precipitation is higher, see **Figure 4**).

[Figure]

*Figure 4 : Evaporation-minus-precipitation (kg/m²/s) for REF in 2015.*

**Minor comments:**

*page 6, line 14: "Spatial correlations in B are modeled by means of an anisotropic*

*Gaussian recursive filter." Please provide the equations on how this filter is built.*

**Answer:** We have referred to the papers below (see references) and do not think it is essential to rewrite the equations.

**References:**

Purser, R. J., W.-S. Wu, D. F. Parrish, and N. M. Roberts, Numerical aspects of the application of recursive filters to variational statistical analysis. Part I: Spatially homogeneous and isotropic Gaussian covariances. Mon. Wea. Rev., 131, 1524-1535, 2003a.

Riishøjgaard, L. P., A direct way of specifying flow-dependent background error correlations for meteorological analysis systems. Tellus, 50A, 42-57, 1998.

Wu, W.-S., R. J. Purser, and D. F. Parrish, Three-dimensional variational analysis with spatially inhomogeneous covariances. Mon. Wea. Rev. 130, 2905-2916, 1992.

**Answer:**

*page 6, line 18: in the definition of d_\xi why is \xi transposed?*

**Answer**: it was a typo mistake, it is changed.

*page 6, line 25: "The Fig. 4 shows the SSS bias ($\xi$ term of the Eq. 1). The patterns are different than the model bias (Fig. 4) and often of opposite sign but have the same magnitude..." Should that not be "Fig. 5" showing the SSS bias?*

*Figure 4: Please add a color-bar on Figure 4.*

**Answer:** Figures and text have been changed for clarity.

*General comment about citation: "Following (Desroziers et al., 2005),..." should be "Following Desroziers et al. (2005),...". Please correct also other similar citation issues.*

**Answer:** corrected

*Salinity is sometimes expressed in PSS (practical salinity scale), for example page 8, line 18 and sometimes in psu (practical salinity unit), for example page 9, line 16.*

*Please make use of the same unit.*

**Answer:** corrected

*page 8, line 26: "Fig. 9 shows the mean and standard deviation of the daily ?? [sic] or monthly differences between the (analyzed) SSS for REF and SMOSexp simulations compared to the SMOS SSS observations (non-debiased). Statistics are computed over the year 2015 for the Tropical Pacific Ocean." Daily or monthly?*

**Answer:** corrected, it is monthly.

*page 10, line 6: "At the equator, there is an acceleration of the Warm Water Pool migration towards the east (Fig.14c) which helps the ocean-atmosphere coupling and thus the triggering of El Niño.". This is not so clear to see from the*

*figure. Can you give the start and end latitude of the Warm Water Pool migration?*

**Answer:** Actualy, because we observe an Eastward acceleration (red color) of the zonal current near the Eastern edge of the warm pool (140°-180°), it induces an acceleration of the Warm Water Pool migration towards the east.

*Hoevmuller -> Hovmöller (or Hovmoeller)*

*define acronym TIW*

**Answer:** All these comments have been taken into account.

[revised manuscript text omitted]
(\boldsymbol{x}) = \frac{1}{2}\,\boldsymbol{x}^T\,\mathbf{B}^{-1}\,\boldsymbol{x} + \frac{1}{2}\,(\mathrm{d}\text{-}\mathbf{H}\,\boldsymbol{x})^T\mathbf{R}^{-1}(\mathrm{d}\text{-}\mathbf{H}\,\boldsymbol{x}) \tag{1}$$

where $d = \;<\textbf{Salinity}_{in-situ}> \; - <\textbf{Salinity}_{model}>\;$ for salinity field

$d$ is the innovation vector of T/S, i.e the mean ($<>$) innovation of in-situ T/S over 1 month in a 1°x1° grid boxes. $\boldsymbol{x}$ is the temperature or salinity in-situ bias to estimate, $\mathbf{B}$ denotes the background error covariance of the 3D bias, $d$ is the innovation vector, $\mathbf{H}$ is the observation operator, $\mathbf{R}$ is the observation covariance error. The vertical grid is a coarse grid (only 23 levels) which is different of the model vertical grid (50 levels). For example, the in-situ innovation at sea surface for T/S is calculated from the average of model and observations between 0 and 11 meters depth. ~~The bias is the minimum of the cost function given by the Eq. 1a, where $x$ is the in-situ bias to estimate, $\mathbf{B}$ is the background error covariance of the 3D bias, d is the innovation vector (it is the mean innovation over 1 month on a 1°x1° grid between 0 and 10 meters depth and the mean is symbolized by $<>$), $\mathbf{H}$ is the observation operator, $\mathbf{R}$ is the observation covariance error. Eq. 1b corresponds to the extra terms to take into account biases in the satellite SSS data.~~

Because temperature and salinity biases are not necessarily correlated at large scales, these two variables are processed separately. Spatial correlations in $\mathbf{B}$ are modeled by means of an anisotropic Gaussian recursive filter (Wu et al., 1992; Riishøjgaard, 1998; Purser et al., 2003). Finally, bias correction of T, S and dynamic height are computed and interpolated on the model grid and applied as tendencies in the model prognostic equations with a 1-month time scale.

**2.4.2 Bias correction scheme for large scale SSS large: SSS from space**

$$J(x,\xi)=\tfrac{1}{2}x^T B^{-1}x+\tfrac{1}{2}(\text{d-}\mathbf{H}\,x)^T R^{-1}(\text{d-}\mathbf{H}\,x) \tag{1a}$$

$$+\tfrac{1}{2}\xi^T B_\xi^{-1}\xi + \tfrac{1}{2}(d_\xi - \mathbf{H}\,\xi)^T R_\xi^{-1}(d_\xi - \mathbf{H}\,\xi) \tag{1b}$$

where

$$d = <SSS_{in-situ}> - <SSS_{model(0.5-10m)}> \text{ and } d_\xi=(<SSS_{SMOS}> -\xi^T) - <SSS_{model(0.5m)}>$$

Earlier attempts to assimilate SSS data have shown the importance of using unbiased satellite SSS data while implementing rigorous quality control in an upstream process (Tranchant et al., 2015). In this study, the bias control of satellite SSS has been modelled by modifying the current T/S bias (in-situ) correction 3D-Var cost function (Eq. 1a). An extra term to take into account biases in the satellite SSS data has been added and denoted $\xi$ in the 3D-Var cost function (Eq.2). The new SSS bias is the minimizer of the cost function given by the Eq. 2.

$$J(x,\xi)=\tfrac{1}{2}x^T \mathbf{B}^{-1}x+\tfrac{1}{2}(\text{d-}\mathbf{H}\,x)^T \mathbf{R}^{-1}(\text{d-}\mathbf{H}\,x) \tag{2}$$
$$+\tfrac{1}{2}\xi^T B_\xi^{-1}\xi + \tfrac{1}{2}(d_\xi - \mathbf{H}\,\xi)^T R_\xi^{-1}(d_\xi - \mathbf{H}\,\xi)$$

where $d_\xi=(<SSS_{SMOS}> -\xi) - <SSS_{model(0.5m)}>$

Here, $d_\xi$ is the innovation of SSS bias at surface, i.e the mean (<>) innovation of SMOS SSS over 1 month on a 1°x1° grid. A new control term for SSS has been added, denoted $\xi$ in the 3DVar cost function (Eq.1b) where $d_\xi$ is the innovation of SSS bias at surface, see eq. 2.

To get an optimal set of parameters (weights, spatial scales and errors), several estimations were performed with data withdrawing. In Figure 3Fig. 4, examples of model salinity bias near the surface ($x$) without (a) (Eq.1) and without (c) (Eq. 2) the SSS bias term ($\xi$) are shown. The patterns are similar except at the equator where the SSS bias (Figure 3b) influences the bias correction of salinity (Figure 3c) with smaller scales. There may also be They have different magnitude due to the addition of the SSS bias. The Fig. 4 shows the SSS bias ($\xi$ term of the Eq. 1). The patterns are different than the model bias (Fig. 4) and often of opposite sign but have the same magnitudeamplitudes are the same. In this example, a persistent large innovations at several depths (11m, 41 m and 79 m) (not shown here) may induce a larger bias of salinity (negative anomaly) at sea surface near 120°W/20°S. The SSS bias from SMOSexp have smaller scales than the model bias.

**2.53.4 SSS observation error**

The Desroziers Diagnostic diagnostic (Desroziers et al., 2005) is commonly used for estimating observation error statistics and is used here to adapt the observation error from the background and analysis residuals calculated in the bias correction, see also (Lellouche et al., 2018). Following (Desroziers et al., (2005), the observation error of the bias $R_\xi$ is optimal when is equal to the statistical expectation of the cross-product between the residual $d_\xi^a$ and the innovation $d_\xi$ of the SSS bias, see Eq. 32.

$$R_\xi = E[d_\xi . d_\xi^a]$$

(32)

Actually, $R_\xi$ is estimated iteratively (n=5) by an iterative boot-strap method computed on a 3°x3° grid.

Five successive analyses are made followed by five estimates of the Desroziers ratio $r_\xi^i$ expressed as Eq. 4 for an analysis i.

$$r_\xi^i = \frac{E[d_\xi.d_\xi^{a_i}]}{R_\xi^i}$$

(4)

From an observation error a priori $R_\xi^o$ and by the successive ratio $r_\xi^{i=1,n}$, we obtain Eq.5:

$$R_\xi = r_\xi^1 \ ... r_\xi^n \ R_\xi^o \quad with \ r_\xi^{i=1,n} = \frac{E[d_\xi.d_\xi^{a_i}]}{R_\xi^{i=1,n}}$$

(5)

The a priori error $R_\xi^o$ is a combination of a zonally varying error, together with an increase over regions with sparse in-situ data and near the coast. This increase varies with the cycle. It means that the SSS bias could not be estimated accurately in the absence of in situ data, and hence will have no impact in the assimilation in those regions void of in situ data. Figure 4  shows an example of the final Desroziers ratio $r_\xi^5$. It illustrates how the fixed zonal error is increased near the equator and   reinforced near central America where in situ data are sparse. There is also a local increase near Samoa (170°W-13°S), probably due to RFI pollution. Several simulations have been done with and without bias correction in order to check the validity of the estimated SSS errors in the data assimilation scheme SAM2.


Finally, for each weekly analysis, the total observation error of satellite SSS (SMOS) prescribed in the data assimilation scheme is the maximum of the above observation error estimated during the bias correction process and the measurements error ($R_{instr.}$) supplied by the data producers (used as a threshold) , see Eq. 46. These measurement error estimates bring smaller scales than can be estimated by the Desroziers diagnostic, see an exemple in Figure 5.

$$R_{Tot} = max(R_\xi , R_{instr.})$$

(6)

(4)


[revised manuscript text omitted]
.enqing, ; Fore, Alexander, F., Yueh, Simon, Y., ; Lee, Tong, L., ; Hayashi, Akiko, H., ; Sanchez, F., ranks, Alejandra; Martinez, A., Justino; King, B.King, Brian;, Baranowski, D.ariusz.;, Validating SMAP SSS with in situ measurements. Remote Sensing of Environment, 200. 326-340. https://doi.org/10.1016/j.rse.2017.08.021, 2017.

Toyoda, T., Fujii, Y., Kuragano T, Matthews J.P., Abe, H., Ebuchi, N., Usui, N., Ogawa, K., Kamachi, M.: Improvements

25   to a Global Ocean Data Assimilation System Through the Incorporation of Aquarius Surface Salinity Data. Q. J. Roy. Meteor. Soc., 141 (692), 2750-2759, doi: 10.1002/qj.2561., 2015.

Tranchant, B., Greiner, E., Llegalloudec, O. And and Letraon P.: Sea Surface Salinity Data Assimilation Improvement in a Global Ocean Forecasting System at 1/4° from SMOS and Aquarius Data, 2nd science SMOS conference, 25-29 May 2015, ESA-ESAC, (near Madrid) Spain, 2015.

30   Vinogradova, N. T., Ponte, R. M., Fukumori, I., and Wang, O.: Estimating Satellite Salinity Errors for Assimilation of Aquarius and SMOS Data into Climate Models. J. Geophys. Res.-Oceans, 119 (8), 4732-4744, doi: 10.1002/2014JC009906, 2014.

Von Schuckmann et al., The Copernicus Marine Service Ocean State Report, Journal of Operational Oceanography, https://doi.org/10.1080/1755876X.2018.1489208, 2018.

Vertenstein, M., Hoffman, F., Oleson, K., and Levis, S.: Community Land Model version 3.0 (CLM3.0) user's guide. Technical report, National Center for Atmospheric Research, 2004.

Wolter, K. and Timlin, M. S. : Measuring the strength of ENSO events – how does 1997/98 rank?, Weather, 53, 315–324, 1998.

5  Wu, W.-S., Purser, R. J., and D. F. Parrish: Three-dimensional variational analysis with spatially inhomogeneous covariances. Mon. Wea. Rev. 130, 2905-2916, 1992

Yin, X., J. Boutin, J., G. Reverdin, G., T. Lee, T., S. Arnault, S., and N. Martin, N., : SMOS Sea Surface Salinity signals of tropical instability waves, J. Geophys. Res. Oceans, 119, 7811–7826, doi:10.1002/2014JC009960, 2014.

[Figure]

[Figure]

**Figure 1**: SSS anomalies (pss) in 2014 (top) and 2015 (bottom): mean salinity difference [(model (control run) – the World Ocean Atlas (WOA) 2013)].

[Figure]

(a)

Figure 2: Representativity error of in-situ SSS (R$_{rep.}$) over the Tropical Pacific used in the data assimilation system.

[Figure]

(b)

**Figure 2: Representativity error of in-situ SSS ($R_{repr.}$)** **(a) and**  **used in the data assimilation system for the week 20-27 January 2016.**

[Figure]

(a)

[Figure]

(b)

**Figure 4: Example of model salinity bias near the surface (x, see Eq. 1a) calculated with the bias correction scheme inferred from in-situ data between 0 and 10 m depth only (a) and with the SSS term (ξ, see Eq. 1b) from SMOS data (b) averaged over 1 month in the Tropical Pacific (week 20-27 January 2016).**

[Figure]

(b)

(c)

**Figure 3: Example of model salinity bias (x) near the surface (Eq. 1a) calculated from in-situ data between 0 and 10 m depth only (a), of SSS bias (ξ) (Eq. 1b) calculated from in-situ data between 0 and 10 m depth and SMOS SSS (b) an salinity bias (x) (Eq. 1a +Eq. 1b) from in-situ data between 0 and 10 m and SMOS SSS (c) in the Tropical Pacific (week 20-27 January 2016).**

**Figure 5: Example of SSS bias ξ calculated with the bias correction scheme inferred from in-situ data between 0 and 10 m and SMOS SSS averaged over 1 month in the Tropical Pacific (week 20-27 January 2016).**

[Figure]

**Figure 4: Example of Desroziers ratio (3°X3°) (see Eq. 4) estimated and applied to the a-priori error (bottom). (week 20-27 January 2016)**

[Figure]

**Figure 5: Example of SSS error (Eq. 5) of SMOS over the Tropical Pacific and used in the data assimilation system for the week 20-27 January 2016.**

The margin boxes: "Mis en forme : Justifié", "Mis en forme : Élevé", "Mis en forme : Élevé, Police :Gras", "Mis en forme : Élevé", "Mis en forme : Police par défaut"

[Figure]

(a)

[Figure]

(b)

[Figure]

(c)

**Figure 6Figure 8:** **RMSE of SSS with respect to** SMOS SSSdata (solid lines) **and RMSE of** in situ **salinity** observations **near 65 meter depth** with respect to in situ salinity data (dashed lines)**,** for the reference simulation (solid line)in the 1-6 day forecast fields in REF (black lines) **and** the SMOSexp simulation (dashed red line) in**,** over **the global domain (**topa**),** over **the Tropical Pacific (**middleb**) and in the Niño3.4 region (**bottomc**). RMSEs are evaluated for each weak and the mean $RMSE$ of the in-situ salinity are denoted in the legend. The regions used here have south-west and north-east corners defined as: Tropical Pacific [30°S, 120°E] to [30°N,70°W]; Niño3.4 [5°S, 170°W] to [5°N, 120°W].**The mean number (weekly) of observations are mentionned.**

[Figure]

[Figure]

**Figure 7Figure 9:** **Mean differenceStatistics (from daily mean) mean** (top) and **root-mean-square-difference (bottom)standard deviation (bottom)** of **monthly mean** SSS **(pss) with respect to the SMOS data** difference (model **minus** -SMOS) observation) **in the analysis fields in** for **REF (left) and SMOSexp (right) experiments** onon 2015 **year** year.

[Figure]

**Figure 8Figure 10: Average salinity RMSE (pss) compared to all in situ measurements (left) over the period 1st Jan 2014 to 2st Mar 2016 in global domain for the *REF* (green line) and *SMOSexp* (red line) experiments as a function of depth over the top 50 m. The corresponding percentage of RMSE difference of all in situ salinity measurements between REF and SMOSexp experiments (right) (positive difference implies a reduction in RMSE by the SSS assimilation).**

[Figure]

[Figure]

**Figure 9Figure 11: Mean October 2015 SSS estimation from the REF experiment (top, left), the SMOSexp experiment (top, right), the SMOS SSS measurements (left, bottom) and annual mean difference (2015) between the SMOSexp and REF experiment (bottom, right). The isohaline 34.8 pss is the (black solid line) is represented.**

[Figure]

**Figure 10Figure 12: Vertical section along the equator of the mean model salinity difference between the SMOSexp and REF experiments for the year 2015.**

[Figure]

**Figure 11Figure 13: Ho­evm­öuller of SSS at 5°N for the REF (left) and SMOSexp (middle) and SMOS data (right)**

**Figure 12Figure 14: Ho­evm­öuller ofdifferences in SSS (left), SST (middle) at 5°N and sea surface zonal velocity (U) (right) at the equator between the SMOSexp and the REF experiment in 2015.**

[Figure]

**Figure 13:** **Hovmöller of Barrier Layer Thickness (BLT) at 5°N for the REF experiment in (left) and for the SMOSexp (right) experiment in 2015.**

[Figure]

[Figure]

[Figure]

**Figure 14Figure 15:** Hoevmöuller ofof SSH (left) and filtered SSH (band-pass filter 28–40 day (33 days) band-passed SSH 28–40 daysanomalies) at 4°N (right) referenced to the temporal annual mean of June-December 2015 for REF experiment ((topleft) and for SMOSexp experiment (rightbottom) during 2015. The propagation speeds of 0.25 and 0.35 m/s (solid lines) are representative of the propagation speed for the 28–40 day bands.

.

[Figure]

(a)

[Figure]

(b)

[Figure]

(c)

**Figure 15Figure 16: Time evolution of the hourly TAO observed salinity (black), the hourly model REF (green), SMOSexp (red) simulations and the assimilated SMOS data (magenta) at three different TAO moorings locations, cold tongue (a) (125°W,7.97°N), warm pool (b) (165°E,4.99°S) and (c) salt front (170°W,4.99°S) from January (2015 to -March 2016). The precipitation rate (blue line) coming from the atmospheric ECMWF forcing is superimposed**

[Figure]

**Figure 16Figure 17. Difference in model salinity RMSE (pss) at 45 m depth calculated against the 1 m depth TAO mooring salinity values (*REF –* SMOSexp _- REF) _calculated over the period 1^st Jan 2015 to 16^th March 2016 (negative/positive difference implies a reduction/increase in RMSE by the SMOS assimilation). Moorings are only included if they have more than 1 week of measurements during the period.**

[Figure]

(a)

Figure 18: Ship routes with TSG salinity observations (PSS)

[Figure]

**Figure 17Figure 19: Ship route of the Matisse with TSG salinity observations (PSS) (a) and TSG Salinity observations compared toand near sea surface salinity analysis (b,c) from the OSEs along the Matisse ship track (LEGOS) (red line= observations, dashed line= REF, black solid line = SMOSexp). A zoom from the orange rectangle of (b) is shown in (c).**

| Instrumental errors ($R_{inst.}$) | |
|---|---|
| Altimetry | |
| JASON2, ALTIKA/SARAL | 2 cm |
| HAIYANG-2A | 4 cm |

| SST | |
|---|---|
| OSTIA L4 | 0.5°C |
| In-situ at sea surface | |
| XBT, moorings, Argo floats, sea mammals | 0.03°C and 0.0075 pss |

**Table 1: Instrumental errors used for the current operational** systemnetwork.

| Experiment name | Period | Assimilated observations | SSS product |
|---|---|---|---|
| Reference (REF) or control run | Jan 2014- March 2016 | Regular observation dataCurrent networks without satellite SSS. | No SSS assimilation |
| SMOSexp | Jan 2014 - March 2016 | Current networkRegular observation data plus SMOS satellite SSS observations. | 4-day 0.25°x0.25° SMOS data from LOCEAN (L3-Debiased-Locean-v2) |

**Table 2: Experiment descriptions.**

| Regions [south-west to north-east corners] | Percentage of RMSE difference of SSS when SMOS SSS is assimilated and mean number of observations | | | |
| --- | --- | --- | --- | --- |
| | SMOS SSS | | In situ salinity near 6 m depth | |
| | % | Mean number of obs./week | % | Mean number of obs./week |
| Global ocean | 24 % | 372,000 | 4.7 % | 1500 |
| Tropical Pacific [30° S, 120° E] to [30° N,70° W] | 26 % | 165,000 | 7.9 % | 500 |
| Niño 3.4 [5°S, 170°W] to [5°N,120°W] | 23 % | 9,500 | 4.8 % | 36 |
| Niño 4 [5°S, 160°E] to [5°N,150°W] | 22 % | 9,500 | 6.7 % | 38 |
| Niño 3 [5°S, 150°W] to [5°N,90°W] | 25 % | 11,400 | 3.3 % | 57 |
| North Tropical Pacific [8°N, 160°E] to [20°N,100°W] | 30 % | 22,300 | 10 % | 33 |
| South Tropical Pacific [20°S, 160°E] to [8°S,90°W] | 24 % | 24,000 | 6.6 % | 64 |

**Table 3: Percentage of RMSE difference of SSS for SMOS and for in-situ salinity at 6 m depth in different regions. The average number of SSS data assimilated per week is also indicated.**

---

## Referee Report (RR1)

The second Review of "Data assimilation of SMOS observations into the Mercator Ocean operational system: focus on the Nino 2015 event" by B. Benoit et al. (os-2018-113).

Although the manuscript is much improved by the region, I think that the current draft needs some confirmations of the correctness and modifications before accepting for the publication. So, I suggest to return the manuscript to authors in order to make another revision.

**General comments**

- 1. I still doubt of the correctness of the cost function, that is, the equation 2. Authors should consider the specific comment 30 and check the correctness again. If the current equation is correct, a suitable explanation is required.
- 2. The explanation of the Desroxiers's method can also be improved. See the specific comment 37.
- 3. The explanation of Figure 16 is not updated, and not consistent with the current figure. See the specific comment 65.
- 4. I feel the English can be still improved. I suggest many modifications in the specific comments.

**Specific Comments**

- 1. P1, L18: "data sets" can be replaced by "datasets".
- 2. P2, L2 "from space make possible": Please modify to "from space makes possible".
- 3. P2, L13 "Barrier Layer": please modify to "barrier layer". It is not necessary to use capital letters. The same mistake can be found in other places.
- 4. P2, L20, "Moreover": I suggest to replace it by "However".
- 5. P2, L22: I suggest to remove "a pattern".
- 6. P2, L23: ", see for example" can be removed.
- 7. P2, L26: "Figure 11" should be "Figure 1".
- 8. P3, L1: "SST, SLA and in situ observations are assimilated as currently done in the operational systems, see Martin et al., (2018).": This sentence should be removed since it is redundant with the sentences in L9-12.
- P3, L7, "it is as strong as the 1997 one, see section 2.6 in (Von Schuckmann et al., 2018).": I suggest to modify this part to "it is as strong as the one in 1997 (Von Schuckmann et al., 2018)."
- 10. P3, L9, "(e.g. NINO3.4 for the Nino3.4 region in the central Pacific, see Barnston et

al., (1997)).": This can be modified to "(e.g., NINO3.4 index is defined as the averaged SST temperature over the area in 5S-5N, 120-170W; see Barnston et al., 1997)."

- 11. 3. P3, L10, "Modoki El Nino": Please modify to "El Nino Modoki".
- 12. P3, L12: I suggest to add "and OSEs" after "the REF simulations".
- 13. P3, L15: ", see Fujii et al., (2015)" can be changed to "(Fujii et al., 2015)".
- 14. P3, L15: "I suggest to replace "Even if" by "Although".
- 15. P3, L28, "The signal to noise ratio is still not high today, thus retrieval algorithms": I suggest to replace the part by "Since the signal to noise ratio is still not high today, retrieval algorithms".
- 16. P4, L3: I think it is better to replace "discussed in section 3" to "presented in section 3".
- 17. P4, L16, "Due to large known biases in precipitation": I suggest to modify the part to "Because there are large known biases in precipitation".
- P4, L21, "instead of Dai and Trenberth (2002).": This part should be omitted because Dai and Trenberth (2002) is not used nor explained.
- 19. P4, L26, "be found in (Lellouche et al., 2018).": Please modify to "be found in Lellouche et al. (2018).".
- 20. P5, L6: I think in-situ observations are used in OSTIA. So it will be double-count if the system also assimilates in-situ SST data directly.
- 21. P5, L17: I do not understand what "18-day products" means.
- 22. P5, L22: It is better to explain the length of the assimilation cycle and the analysis date before this sentence.
- 23. P6, L8, "an incremental analysis update, see (Bloom et al., 1996; Benkiran and Greiner 2008).": I suggest to modify the part to "an incremental analysis update (Bloom et al., 1996; Benkiran and Greiner 2008)."
- 24. P6, L22: I think there are also biases between model and data for sea level anomalies.
- 25. P6, equation (1): Please change "d" to the bold letter.
- 26. P6, L2: Please put a period after "for salinity field".
- 27. P6, L3: Please insert "Here, " before "d is the ...", and remove "(<>)" and add after the sentence "Salinityin-situ and Salinitymodel denote salinity values of in-situ data and in the model, and < • > indicates the mean."
- 28. P6, L10, "bias correction of T, S and dynamic height are computed": The bias correction of the dynamic height is used in the analysis? If so, I think it is inconsistent with that at P6, L22.
- 29. 3. P7, L10, "in the 3D-Var cost function (Eq.2).": I suggest to modify this part to "in the following 3D-Var cost function:" and to move the flowing sentence before "In Figure 3" at

L18.

30. P7, Equation (2): I still think there is something wrong in the equation (2). First, **H** in the fourth term of the righthand side must be different from the one in the second term. So, I recommend to substitute **H** in the fourth term with  $\mathbf{H}_{\xi}$ . In addition, I think  $\mathbf{d}_{\xi}$  must be defined as follows:

 $d_{\xi} = < \textit{SSS}_{\textit{SMOS}} > - (< \textit{SSS}_{\textit{model}} > + H_{\xi 2} x)$  .

In this form, the bias is corrected for the model values using **x**, while the raw SMOS values are used for the innovation because the bias correction of SMOS,  $\xi$ , will be applied to the raw values of SMOS. Here,  $\mathbf{H}_{\xi_2}$  is the linear operator which interpolates **x** to the positions of SMOS observations. If this is correct, the equation (3) is satisfied. Also, authors imply that **x** and  $\xi$  cannot be determined independently in the explanation of Fig. 3. It means the **x** is included in the definition of  $\mathbf{d}_{\xi}$ , because otherwise **x** can be determined independently. I would like to ask authors to check the equation again, and if I am not correct, they should carefully explain the equation so that I can understand it. You must also denote that what  $\mathbf{B}_{\xi}$ ,  $\mathbf{R}_{\xi}$ ,  $SSS_{SMOS}$ , and  $SSS_{model}$  represent in the text. (Definitions of all mathematical symbols must be defined in the text.)

- 31. P7, L16: "(<>)" is not necessary here and should be removed.
- 32. P7, L18, "In Figure 3, examples of salinity bias near the surface (x) without (a) (Eq.1) and with (c) (Eq. 2) the SSS bias term (ξ) are shown.": I suggest to modify the sentence to "Figure 3a and c show examples of the model salinity bias, x, near the surface without and with the estimation of the bias of SMOS data, ξ."
- 33. P7, L19, "where the SSS bias (Figure 3b) influences the bias correction of salinity (Figure 3c) with smaller scales.": I suggest to modify the part to "where the estimated bias of SMOS data (Figure 3b) influences the estimated model salinity bias (Figure 3c) with smaller scales."
- 34. P7, L20, "There may also be opposite sign but amplitudes are the same.": I do not know what authors intend to explain by this sentence.
- 35. P8, L3, "in the bias correction, see also (Lellouche et al., 2018).": I suggest to modify the part to "in the bias correction (see also Lellouche et al., 2018)."
- 36. P8, L5: ", see Eq. 3" should be removed.
- 37. The explanation of the method based on Desroxiers et al. (2005) is improved, but still not clear. In my understanding, the system assumes that  $\mathbf{R}_{\xi}$  is diagonal and calculate the optimal factor of each diagonal element separately. If so, they should explain about that at first. Then, they forgot to define  $\mathbf{d}_{\xi}^{a_i}$  and  $\mathbf{R}_{\xi}^i$  (i = 1, 2, ..., n) and probably the equation (4) should be

$$r_{\xi}^{i} = \frac{E[d_{\xi}d_{\xi}^{a_{i}}]}{R_{\xi}^{i-1}}$$
.

Here,  $d_{\xi} d_{\xi}^{a_i}$ , and  $R_{\xi}^{a_i}$  are an element of  $\mathbf{d}_{\xi} \mathbf{d}_{\xi}^{a_i}$ , and  $\mathbf{R}_{\xi}^{a_i}$ . Please note that the number on the right shoulder of R is *i*-1. I recommend authors to improve the description with considering the comments above.

- 38. P8, L12: This sentence says that the prior error have already increased over regions with sparse in-situ data and near the coast. Is it correct?
- 39. P8, Figure 4: It is more appropriate to show the product  $r_{\xi}^{1} r_{\xi}^{2} r_{\xi}^{3} r_{\xi}^{4} r_{\xi}^{5}$  instead of  $r_{\xi}^{5}$ .
- 40. P8, L21: I suggest to remove ", see Eq.6.", and to move the following sentence after the equation with removing the last part ", see an exemple in Figure 5".
- 41. P9, L18: It is better to mention that the forecasted field is mostly independent of the reference data because those data have not been assimilated yet.
- 42. P9, L19 "Figure 6 shows the time-series of Root-Mean-Square Errors (RMSEs) between the model near-surface salinity (6 m depth) compared to in situ observations (dotted lines) and between the model SSS (0.5 m depth) compared to the bias-corrected SMOS SSS (solid lines) for both simulations (REF in black, SMOSexp in red).": I suggest to modify the sentence to "Figure 6 shows the time-series of Root-Mean-Square Errors (RMSEs) of the model near-surface salinity at 6 m depth with respect to in situ observations (dotted lines) and of the model SSS (0.5 m depth) with respect to the biascorrected SMOS SSS (solid lines) for both simulations (REF in black, SMOSexp in red)."
- 43. P10, L5 "is very weak": I suggest to replace "weak" to "small".
- 44. P10, L11: "Figure 11" should be "Figure 1".
- 45. P10, l23: Please modify "3.1.1" to "3.1.2".
- 46. P20, L28: "in agreement with (Kidd et al., 2013)" can be modified to "in agreement with Kidd et al. (2013)",
- 47. P20, L32, "changes to the SMOS SSS data assimilation": I suggest to replace the part to "changes brought by SMOS SSS data assimilation"
- 48. P20, L33, "The largest magnitudes (saltier)": I suggest to modify it to "The largest highsalinity anomaly"
- 49. P11, L8: "Figure 11" should be "Figure 1".
- 50. P11, L9 "Both the REF and SMOSexp simulations represent the decrease in time of the salinity peaking in fall 2015 at this latitude, for the longitude between 160°E 10 and 120°W.": I suggest to modify the sentence to "Both the REF and SMOSexp simulations represent the decrease of the salinity in fall 2015 between 160°E and 120°W."
- 51. P11, L11, "salinity anomaly is lower": I think "salinity anomaly is smaller" is better.
- 52. P11, L15, "it is not the case when we are looking at model fields time changes": I suggest

to modify the part to "it is not the case when we look at the time-evolution of model fields"

- 53. P11, L16, "Indeed, an impact can be seen on the other surface variables.": This sentence can be omitted, and the next sentence can be started from "Indeed,".
- 54. P11, L20, "but it this effect": Please omit "it".
- 55. P11, L21, "the eastern warm water pool migration": I suggest to modify the part to "the eastward warm water pool migration".
- 56. P11, L23, "Tropical Instability Waves (TIWs), see Figure 14.": I suggest to change this part to "Tropical Instability Which will be shown later."
- 57. P11, L31, ", see Qu et al., (2014)": I suggest to change the part to " (e.g., Wu et al., 2014)".
- 58. P12,L4, "33-day TIWs are": I suggest to change this to "TIWs, which has a 33-day period, are".
- 59. P12, Figure 14: It looks to me that TIWs do not propagate in parallel to the line representing the speed of 0.25 m/s.
- 60. .P12: It is very interesting that assimilating SMOS data enhances the activity of TIWs. It would be nice if some discussion why the activity of TIWs are enhanced by the SMOS data assimilation.
- 61. P12, L29, "between the TAO observations and SMAP/SMOS observations and Argo analysis": I suggest to change the part to "among the TAO, SMAP/SMOS, and Argo analysis".
- 62. P12, l29, "There is an improvement in the cold tongue during the end of summer, in fall 2015 and during the last 2 months of the SMOS simulation (15a) in the region where the data assimilation of SMOS reduces the freshening.": I suggest to modify the sentence to "There is an improvement in the cold tongue during the end of summer, in fall 2015 and during the last 2 months of the SMOS simulation (Figure 15a). The data assimilation of SMOS reduces the freshening."
- 63. P13, L2, "Obviously, the assimilated 4-days SMOS data are smoother but are able to capture the large scale variability.": I am not sure what "4-days" means. I also suggest to change the sentence to "Obviously, the time-series of the assimilated 4-days SMOS data is smoother but able to capture the large scale variability."
- 64. P13, L6, "This also shows that the observation error should not be increased locally depending on the precipitation.": I suggest to modify the sentence to "This also shows that the observation error is not necessarily increased locally depending on the precipitation." I also point out it is difficult to judge whether the observation error is increased by local precipitation or not. At least, I think some of the increase of the observation error is caused by local precipitation.
- 65. P13, Figure 16: The description of the figure is not consistent with the current figure.

Probably the figure is updated but the description is not updated.

- 66. P13, L 15, "This reflects the overestimation of E-P that the data assimilation tends to correct and the SMOSexp experiment is saltier in regions where precipitation is higher.": I suggest to modify the sentence to "This reflects the tendency that the SMOS data assimilation reduce the low salinity biases by mitigating the overestimation of E-P in the regions of large precipitation."
- 67. P13, L19: please correct "3.2.1" to "3.2.2".
- 68. P14, L10 "for other purpose": Authors should give readers some examples about this.
- 69. P14, L30 "This also reflects that the overestimation of E-P is corrected by data assimilation through salting in regions where precipitations are higher.": I suggest to modify the sentence to "This also reflects that the overestimation of E-P is mitigated by the data assimilation through salting in the regions of large precipitation."
- 70. P15, L3, "But, an impact on SSH have been seen through TIWs which have been reduced (amplitude and propagation speed) and then strengthened in the eastern part of the basin during the last half of the 2015 year.": I suggest to change the sentence to "But, an impact on TIW have been seen through SSH fields. Amplitudes and propagation speed of TIWs are reduced while their activity is enhanced in the eastern part of the basin during the last half of 2015."
- 71. P15, L4 "through a positive feedback.": There is no explanation on the positive feedback.
- 72. The caption of Figure 2: The "and" at the beginning of the second line can be removed.
- 73. The caption of Figure 3: This caption is not modified according to the revision of the main text. In particular, Eq 1a and 1b are not used in the current draft.

---

## Author Response (AR2)

**ANSWERS TO REVIEWER# 1**

**General comments:**

**1.**I still doubt of the correctness of the cost function, that is, the equation 2. Authors should consider the specific comment 30 and check the correctness again. If the current equation is correct, a suitable explanation is required.

**Answer:** This Equation was wrong and a lot of corrections and clarifications have been made.

**2.**The explanation of the Desroziers's method can also be improved. See the specific comment 37.

**Answer:** The description of the method has been improved.

**3.**The explanation of Figure 16 is not updated, and not consistent with the current figure. See the specific comment 65.

**Answer**: No, the explanation is consistent with the Figure 16: "The impact of the SMOS assimilation is contrasted by showing negative (positive) values which indicates that it reduces (increases) the RMSD." When the difference in model salinity RMSE is negative (blue), it indicates that there is a positive impact, the RMSE or RMSD is reduced.

**4.**I feel the English can be still improved. I suggest many modifications in the specific comments.

**Answer**: We took into account all the specific comments.

**Specific Comments:**

**1 to 19**: **Modifications have been made.**

**20. P5, L6**: I think in-situ observations are used in OSTIA. So it will be double-count if the system also assimilates in-situ SST data directly.

**Answer**: In-situ observations are used in the OSTIA analysis products but they are only used for the large scale bias correction (i.e., the differences are analysed on a ~1/4 degree grid with 7 degree background error covariance lengthscale). Consequently, we cannot therefore consider that in-situ observations are assimilated twice into the system.

**21. P5, L17:** I do not understand what "18-day products" means.

**Answer**: as written in http://www.catds.fr/Products/Available-products-from-CEC-OS/L3-Debiased-Locean-v2 , SMOS L3 products used in this study are SSS maps provided every 4 days from 01/2010 to 12/2016 and are derived from a combination of ascending and descending orbits. Debiased SSS are temporally averaged using a slipping Gaussian kernel with a full width at half maximum of 18 days (18 days product).

**22. P5, L22:** It is better to explain the length of the assimilation cycle and the analysis date before this sentence.

**Answer**: We added precisions in this section and add this following sentence in the section 2.3:

"As in the operational ocean forecasting system, we use a weekly assimilation cycle with an analysis date on the fourth day of the week "

**23. P6, L8,** "an incremental analysis update, see (Bloom et al., 1996; Benkiran and Greiner 2008).": I suggest to modify the part to "an incremental analysis update (Bloom et al., 1996; Benkiran and Greiner 2008)."

**Answer**: The suggestion has been taken into account.

**24. P6, L22**: I think there are also biases between model and data for sea level anomalies.

**Answer**: Yes, it was a mistake. There are also biases in the sea level anomalies which should be removed in the mean dynamic topography. The sentence has been changed.

**25-27: Changes have been made**

**28. P6, L10**, "bias correction of T, S and dynamic height are computed": The bias correction of the dynamic height is used in the analysis? If so, I think it is inconsistent with that at P6, L22.

**Answer**: See the correction of the specific comment 25. Consequently, it is true that the bias correction of T, S and dynamic height are computed and interpolated on the model grid and applied as tendencies in the model prognostic equations with a 1-month time scale.

**29. 3. P7, L10,** "in the 3D-Var cost function (Eq.2).": I suggest to modify this part to "in the following 3D-Var cost function:" and to move the flowing sentence before "In Figure 3" at L18.

**Answer**: We modified the sentence but we think that it is the right place because it introduces the Equation 2.

**30. P7, Equation (2):** I still think there is something wrong in the equation (2)….

**Answer**: The reviewer is right, there was a mistake in the equation. We also clarified and explained all terms. However, we prefer to write $d_\xi$ as follows :

$d_\xi$=(**<SSS**$_{SMOS}$ > $-\xi$) $-$ **<SSS**$_{model(0.5m)}$ > because, the first term (**<SSS**$_{SMOS}$ > $-\xi$) denotes the unbiased SMOS SSS.

**31-36: Changes have been made.**

**37**. The explanation of the method based on Desroziers et al. (2005) is improved, but still not clear. In my understanding, the system assumes that $R_\xi$ is diagonal and calculate the optimal factor of each diagonal element separately. If so, they should explain about that at first. Then, they forgot to define $d_\xi^{a_i}$ and $R_\xi^i$ ($i = 1, 2, \dots n$) probably the equation (4) should be

$$r_\xi^i = \frac{E\left[d_\xi . d_\xi^{a_i}\right]}{R_\xi^{i-1}}$$

Here, $d_\xi$ $d_\xi^a$, and $R_\xi^{a_i}$ are an element of $d_\xi$ $d_\xi^a$, and $R_\xi^{a_i}$. Please note that the number on the right shoulder of **R** is i -1. I recommend authors to improve the description with considering the comments above.

**Answer**: Yes, $R_\xi$ is diagonal and the method calculates the optimal factor of each diagonal element separately. We improved the description.

**38. P8, L12**: This sentence says that the prior error has already increased over regions with sparse in-situ data and near the coast. Is it correct?

**Answer**: yes, it is correct, the bias correction cannot work without in-situ data.

39. P8, Figure 4: It is more appropriate to show the product $r_\xi^5 r_\xi^4 r_\xi^3 r_\xi^2 r_\xi^{51}$ instead of $r_\xi^5$ .

**Answer**: yes, it is correct, we changed the sentence and the caption because it is what we showed originally, it was not $r_\xi^5$ as previously mentioned.

**40-58: Changes have been made.**

**59. P12, Figure 14:** It looks to me that TIWs do not propagate in parallel to the line representing the speed of 0.25 m/s.

**Answer**: Yes, it is right, the value has been adjusted to 0.20 m/s, see the new Figure 14.

**60. P12:** It is very interesting that assimilating SMOS data enhances the activity of TIWs. It would be nice if some discussion why the activity of TIWs are enhanced by the SMOS data assimilation.

**Answer**: The goal of this paper is to show that assimilating SMOS SSS may induce some density changes and consequently can play a role on BLT. And we know that a decrease of BLT (decrease of stratification) may induce an increase of the convective mixing which could be also enhanced by TIWs activity. Indeed, with a weaker stratification, the thermocline is deeper and then influence the wave propagation by accelerating it.

Some precisions have been brought in this section and in the discussion.

**61-64: Changes have been made.**

**65. P13, Figure 16:** The description of the figure is not consistent with the current figure. Probably the figure is updated but the description is not updated

**Answer**: See the answer 3 in the general comment.

**66-70: Changes have been made.**

71. P15, L4 "through a positive feedback.": There is no explanation on the positive feedback.

**Answer**: see changes in the text related to specific comment 60. We removed the term positive feedback. It is more appropriate to write that mixing can be modified by TIWs and SSS data assimilation but TIWs can also be modified by the change of stratification.

**72-73: Changes have been made.**

**ANSWERS TO REVIEWER# 2**

**Minor issues:**

**\* page 7: 2.4.2 Bias correction scheme for large scale SSS large: SSS from space drop the last "large"**

**Answer:** Change has been made.

**\* equation 2: It seems that the cost function is decoupled between x and \eta? If there is a coupling between both the terms in x and \eta, please make such coupling clear from the equation and if not, please state so in the text for clarification.**

**Answer:** There was a mistake in the equation 2. We changed it and explained all terms of this equation.

[revised manuscript text omitted]